# Research

medicinal chemistry

spirooxindoles, tetrahydroisoquinolines, molecular hybridization, cancer

**Author for correspondence:**
Simon M. N. Efange
e-mail: efange.mbua@ubuea.cm

This article has been edited by the Royal Society of Chemistry, including the commissioning, peer review process and editorial aspects up to the point of acceptance.

# 3′,4′-Dihydro-2′H-spiro[indoline-3,1′-isoquinolin]-2-ones as potential anti-cancer agents: synthesis and preliminary screening

## Maloba M. M. Lobe and Simon M. N. Efange

Department of Chemistry, University of Buea, PO Box 63, Buea, Cameroon

Both tetrahydroisoquinolines (THIQs) and oxindoles (OXs) display a broad range of biological activities including anti-cancer activity, and are therefore recognized as two privileged scaffolds in drug discovery. In the present study, 24 3′,4′-dihydro-2′H-spiro[indoline-3,1′-isoquinolin]-2-ones, designed as molecular hybrids of THIQ and OX, were synthesized and screened *in vitro* against 59 cell lines in the NCI-60 screen. Twenty compounds displayed weak to moderate inhibition of cell proliferation; among them, three compounds displayed at least 50% inhibition of cell proliferation. The compounds appeared to target primarily renal cell cancer lines; however, leukaemia, melanoma, non-small cell lung cancer, prostate, ovarian and even breast cancer cell lines were also affected. Therefore, this class of spirooxindoles may provide useful leads in the search for new anti-cancer agents.

## 1. Introduction

The term cancer refers to a group of more than 100 diseases characterized by a common feature of rapid and uncontrolled cell proliferation, invasiveness and metastasis, leading to death. In 2018, worldwide mortality from cancer alone accounted for an estimated 9.6 million deaths, thereby positioning cancer as the number two leading cause of death worldwide [1]. According to statistics from the World Health Organization, five forms of cancer account for the most deaths: lung (1.76 million deaths), liver (782 000 deaths), colorectal (862 000 deaths), stomach (783 000 deaths) and breast (627 000 deaths) [1,2]. In 2010, the total economic cost of cancer was estimated at 1.16 trillion US dollars [1]. When combined with the human cost of the disease, and the increasing incidence of the disease(s), there

is no disputing that cancer constitutes a major public health problem worldwide that requires continued attention.

The clinical management of cancer relies on four main pillars: surgery, hormone therapy, radiotherapy and chemotherapy [3]. While these are often used in combination, there is a heavy reliance on chemotherapy. However, the effectiveness of many anti-cancer agents is compromised by chemoresistance. The toxic side effects of some of these drugs also limit their clinical utility. Consequently, there is a continuing need to develop new agents, particularly those with novel mechanisms of action that can be deployed in the clinic.

Privileged scaffolds, as defined by Evans *et al.* [4], are molecular frameworks that occur in numerous biologically active low molecular weight compounds. The recurrence of these molecular frameworks in biologically active molecules has led to the idea that these scaffolds are recognized by several molecular targets. Consequently, privileged scaffolds are frequently employed as starting points in drug discovery efforts, and the combination of two or more of such scaffolds, termed molecular hybridization [5], has therefore been adopted as a useful strategy for designing biologically active molecules. The present work describes the hybridization of two privileged scaffolds, tetrahydroisoquinoline (THIQ) and oxindole (OX), to form a new class of potential anti-proliferative agents.

The THIQs display a wide range of biological activities including central nervous system, cardiovascular, anti-tumour, antibacterial, antimicrobial, antitubercular, anti-viral, antifungal, antileishmanial, antitrypanosomal and antiplasmodial activities [6–16]. Their anti-tumour activity is attributed to several mechanisms, including induction of apoptosis and increase in caspase-3, -8, -9 and p53 [17]; downregulation of CDK2 and upregulation of the tumour suppressor genes p21, p27 and p53 [18]; inhibition of the MDM2–p53 interaction [19]; reversal of multidrug resistance [20]; inhibition of histone deacetylase [21]; and induction of G2/M cell cycle arrest and mitosis through the disruption of microtubule dynamics [22]. Molecules constructed from the THIQ scaffold may therefore be expected to display anti-proliferative activity.

Naturally occurring (figure 1) and synthetic OXs have also been found to display a diverse array of biological properties, including antimalarial, antifungal, anticonvulsant, anti-HIV, radical scavenging and antibacterial activities and anti-cancer activities [23–29]. Their anti-cancer activity is attributed to a number of mechanisms such as inhibition of caspase [30,31], p53 induction [32] and inhibition of the tumour suppressor gene p53 [33].

In recent years, spirocycles in general and spirooxindoles in particular have attracted much attention due to their unique structural aspects and their presence in a number of bioactive natural products [34,35]. Naturally occurring and synthetic spirooxindoles display a wide array of pharmacological activities including anti-cancer [36,37], anti-inflammatory [38], antimicrobial [35], antimalarial [39], antitubercular [40], antileishmanial [41], anti-cholinesterase [35] and anti-viral [42], activity, and therefore continue to attract interest as leads in drug discovery. Among naturally occurring spirooxindoles, the spiro[pyrrolidine-2:3-oxindole] scaffold (found in elacomine, horsfiline and rhynchophylline) appears quite frequently. This scaffold which may be regarded as arising from the intramolecular Mannich reaction between 3-(2-aminoethyl)indolin-2-one, an oxidation product of tryptamine, with an aldehyde has attracted considerable interest and provided many synthetic spirooxindoles [36]. However, the related class of spirooxindoles that arise from the Pictet–Spengler reaction of tryptamine with isatin (a naturally occurring compound) has been largely ignored save for work on the antimalarial agent NITD609 [39,43]. By extension, spirooxindoles, specifically 3′,4′-dihydro-2′H-spiro[indoline-3,1′-isoquinoline]-2-ones, that are obtained from the Pictet–Spengler type reaction between phenethylamines, another class of naturally occurring compounds, with isatin have also received scant attention. Formally, the latter spirooxindoles can be regarded as molecular hybrids of THIQ and OX arising from the Pictet–Spengler type reaction between a substituted phenethylamine and OX. In view of the anti-proliferative activity of both THIQs and OXs (including spirooxindoles), we proposed that this particular class of spirofused molecular hybrids of THIQ and OX would display anti-proliferative activity. A review of the literature at the beginning of the study revealed that the 3′,4′-dihydro-2′H-spiro[indoline-3,1′-isoquinolin]-2-one scaffold had been mentioned merely as a by-product of the reaction between isatin and some amines [44] and in a subsequent study of the kinetic and thermodynamic control of the Pictet–Spengler reaction of phenethylamines and isatin [45]. The present study thus focuses on the synthesis of 3′,4′-dihydro-2′H-spiro[indoline-3,1′-isoquinolin]-2-ones as molecular hybrids of THIQ and OX, and the evaluation of their anti-proliferative activity (figure 2).

**Figure 1.** Examples of naturally occurring spirooxindoles.

## 2. Results and discussion

The target compounds were synthesized by the Pictet–Spengler reaction of a substituted phenethylamine with isatin or a substituted isatin. The precursors, 3,4-dimethoxyphenethylamine and 3,4,5-trimethoxyphenethylamine were prepared as previously described for related amphetamine analogues [46], while 3-hydroxyphenethylamine was obtained from the demethylation of commercially available 3-methoxyphenethylamine [47]. The N-alkylation of isatin and its analogues has also been reported [48]. For the synthesis of the hydroxylated compounds (**1a–q**, **2a–q** and **3a,b**), the phenolic Pictet–Spengler reaction, earlier described by Kametani *et al.* [47], was found to be quite useful as it proceeded smoothly under relatively mild conditions of heating in ethanol. This reaction generally provided two positional isomers, the 6-hydroxy (**2a–q**) and 8-hydroxy (**1a–q**) compounds, in variable ratios. In some instances, only trace amounts of the minor isomer were detected; therefore, only one isomer was isolated. The regioisomers were easily separated by medium pressure liquid chromatography or vacuum liquid chromatography on a silica gel column. To prepare the methoxyl-substituted

a: $R_1 = R_2 = R_3 = H$
b: $R_1 = Cl$; $R_2 = R_3 = H$
c: $R_1 = R_2 = Br$; $R_3 = H$
d: $R_1 = R_2 = H$; $R_3 = 4$-fluorobenzyl
e: $R_1 = R_2 = H$; $R_3 = 4$-chlorobenzyl
f: $R_1 = R_2 = H$; $R_3 = 4$-bromobenzyl
g: $R_1 = R_2 = H$; $R_3 = 3,4$-dichlorobenzyl
h: $R_1 = R_2 = H$; $R_3 = 4$-methylbenzyl
i: $R_1 = R_2 = H$; $R_3 = 2$-nitrobenzyl

j: $R_1 = R_2 = H$; $R_3 = (2$-naphthyl)methyl
k: $R_1 = Cl$; $R_2 = H$; $R_3 = 4$-fluorobenzyl
l: $R_1 = Cl$; $R_2 = H$; $R_3 = 4$-chlorobenzyl
m: $R_1 = Cl$; $R_2 = H$; $R_3 = 4$-bromobenzyl
n: $R_1 = Cl$; $R_2 = H$; $R_3 = 3,4$-dichlorobenzyl
o: $R_1 = Cl$; $R_2 = H$; $R_3 = 4$-methylbenzyl
p: $R_1 = Cl$; $R_2 = H$; $R_3 = 2$-nitrobenzyl
q: $R_1 = Cl$; $R_2 = H$; $R_3 = (2$-naphthyl)methyl

a: $R_1 = R_2 = R_3 = H$
b: $R_1 = Cl$; $R_2 = R_3 = H$
c: $R_1 = R_2 = Br$; $R_3 = H$
d: $R_1 = R_2 = H$; $R_3 = 4$-fluorobenzyl
e: $R_1 = R_2 = H$; $R_3 = 4$-methylbenzyl
f: $R_1 = Cl$; $R_2 = H$; $R_3 = 4$-bromobenzyl

**Figure 2.** Structures of target compounds of the study.

spirooxindoles (compound series **4**, **5** and **6**), the Pictet–Spengler reaction of the methoxyphenethylamines with the corresponding isatins was best carried out in polyphosphoric acid at 100°C [49]. The yields were moderate to good. Under these reaction conditions, only a single regioisomer was obtained from the Pictet–Spengler reaction of 3-methoxphenethylamine with the isatins. All the compounds were converted to the corresponding hydrochlorides in methanolic HCl, recrystallized and submitted for biological testing.

The target compounds were submitted for anti-cancer screening under the NCI-60 programme of the National Cancer Institute, Bethesda, MD, USA (https://dtp.cancer.gov/discovery_development/nci-60/methodology.htm). In this protocol, test compounds are screened at a concentration of 10 µM against 59 cancer cell lines. Of 24 compounds tested, 20 were found to display weak to moderate anti-proliferative activity for at least one cancer cell line (table 1).

Three compounds (**3b**, **4d**, **5f**) inhibited cell proliferation by at least 50% and compound **5f** appeared to be the most active across a number of cell lines. Furthermore, renal cell cancer appeared to be the most susceptible to this class of compounds as up to 10 analogues displayed some anti-proliferative activity on this cancer cell line. Three compounds exerted anti-proliferative activity on the PC-3 cancer cell line, while the ovarian and breast cancer cell lines were inhibited by two and one compound, respectively.

Among the spirooxindoles showing evidence of anti-proliferative activity, analogues containing the 6,7-dimethoxy-THIQ fragment (series **5**) appeared with the highest frequency while those containing the 6-methoxy-THIQ fragment (series **3**) appeared with the lowest frequency. Therefore, the 6,7-dimethoxy-THIQ fragment appears to be the preferred fragment for subsequent optimization efforts of this hybrid scaffold. Of the 20 compounds showing notable anti-proliferative activity, nine (45%)

**Table 1.** Inhibition of cancer cell growth by target spirooxindoles (concentration 10 µM).

| cancer type | compound (% inhibition of cell growth) |
|---|---|
| leukaemia (K-562) | **5f** (44.6) |
| leukaemia (MOLT-4) | **5f** (43.0) |
| leukaemia (RPMI-8226) | **5d** (34.6); **5e** (47.8); **5f** (53.6) |
| leukaemia (SR) | **5f** (37.7) |
| non-small cell lung cancer (HOP-92) | **1e** (30.1); **2h** (32.0) |
| colon cancer (HCT-116) | **5b** (31.5) |
| central nervous system (SNB-75) | **4c** (29.4); **5b** (31.1) |
| melanoma (SK-MEL-5) | **5f** (43.3) |
| melanoma (UACC-62) | **5d** (30.3); **5e** (33.5); **5f** (45.0) |
| ovarian (IGROV1) | **1c** (37.2) |
| ovarian (OVCAR-4) | **5f** (39.3) |
| renal (A498) | **3b** (50.2); **4b** (29.6); **4c** (33.0); **4d** (52.9); **4e** (46.1); **5a** (36.7); **5c** (35.7) |
| renal (CAKI-1) | **2c** (35.5); **4e** (39.0); **5f** (30.1) |
| renal (UO-31) | **1c** (35.3); **1e** (35.2); **2 g** (32.4); **4c** (37.9); **4d** (41.9); **5d** (36.6); **6b** (31.5); **6c** (49.4); **6d** (35.8); **6e** (35.9) |
| prostate (PC-3) | **5d** (33.8); **5e** (42.2); **5f** (52.1) |
| breast (MDA-MB-468) | **5f** (35.7) |

were unsubstituted at the nitrogen atom of the OX fragment, while the rest featured a substituted benzyl fragment at this position. Therefore, subsequent optimization efforts may concentrate on the synthesis of either N-unsubstituted or N-substituted analogues. In the light of these findings, we conclude that the 3′,4′-dihydro-2′H-spiro[indoline-3,1′-isoquinolin]-2-one scaffold does possess anti-proliferative activity.

A recent investigation of the isomeric 2′,4′-dihydro-1′H-spiro[indoline-3,3′-isoquinolin]-2-ones, published while the current study was underway, found that the latter compounds display anti-cancer activity on colon cancer [50]. The most potent compound of that study was found to inhibit Ras-GTP and thereby suppress downstream signalling by MAPK, PI3 K-Akt and Wnt, ultimately resulting in mitochondrial apoptosis [50]. Therefore, while the mode of action of the target 3′,4′-dihydro-2′H-spiro[indoline-3,1′-isoquinolin]-2-ones of the present study is unknown, it does appear that spirofused molecular hybrids of THIQ and OX may be potentially useful leads for anti-cancer drug discovery.

Salsolinol [51] and many other naturally occurring THIQs arise from the Pictet–Spengler type reaction of a biogenic amine with an aldehyde or ketone. The design of the target compounds of this study therefore takes advantage of an endogenous pathway (biomimetic approach) that provides broad flexibility in the construction of compound libraries. Further investigation is underway to optimize the anti-proliferative activity of the target 3′,4′-dihydro-2′H-spiro[indoline-3,1′-isoquinolin]-2-ones against cell lines of interest and to elucidate their mechanism(s) of action.

# 3. Experimental procedure

## 3.1. General

All chemicals were purchased from Sigma Aldrich Chemicals Company, St Louis, MO, USA, and were used as supplied. All solvents were reagent grade. Where necessary, solvents and starting materials were purified using standard procedures. Solvent removal was performed *in vacuo* using a Buchi rotatory evaporator at temperatures not greater than 60°C. Melting points were measured using the Mel-Temp II apparatus with the use of open capillaries and are uncorrected.

The progress of all reactions was monitored using thin layer chromatography (TLC) on aluminium-backed sheets of Sigma Aldrich silica gel 60 $F_{254}$ plates. Compounds were visualized under UV light at

254 nm or in an iodine chamber. Chromatographic purification of compounds was carried out by medium pressure liquid chromatography using silica gel 60–400 mesh. The solvent mixtures used in specific chromatographic runs are indicated where necessary.

High-resolution Fourier transform mass spectrometry electrospray ionization (FTMS-ESI) mass spectra were carried out on an LTQ Orbitrap XL mass spectrometer from Thermo Fisher Scientific (Bremen, Germany). A heated electrospray interface (H-ESI) was operated for ionization of the molecules at a spray voltage of 5 kV, whereas capillary voltage and tube lens voltages were adjusted to 20 and 100 V, respectively. The vaporizer temperature was set at 250°C and the ion transfer capillary temperature to 200°C. Measurements were carried out in the positive ion mode in a mass range of $m/z$ 100–600 at a mass resolution of 60 000 at $m/z$ 200. MS/MS experiments were performed using argon as collision gas in collision-induced dissociation (CID) mode, collision energies were measured at 15, 25 and 35 eV.

Nuclear magnetic resonance spectra were obtained using a Brucker Avance III spectrometer operating at 600 MHz ($H^1$) and 150 MHz ($^{13}C$). Spectra were recorded in deuterated solvents as indicated in brackets and referenced to residual solvent signals. Chemical shifts ($\delta$) were measured in parts per million (ppm). Hydrogen and carbon peak assignments were performed using gradient correlation spectroscopy (gCOSY), heteronuclear single quantum correlation (gHSQC) spectroscopy and heteronuclear multiple bond correlation (gHMBC) techniques. For most of the compounds, signals of protons attached to oxygen and nitrogen were not observed and were attributed to exchange with solvent protons. Signal multiplicities are reported as singlet (s), doublet (d), doublet of doublets (dd), doublet of triplets (dt), triplet (t), triplet of doublets (td) and multiplet (m). Coupling constants ($J$) are reported in Hertz units.

Biological screening: The target compounds were tested in the NCI-60 screen following the published protocol (https://dtp.cancer.gov/discovery_development/nci-60/methodology.htm).

## 3.2. Synthesis of 3-hydroxyphenethylamine (method A)

This compound was synthesized following the method of Ngo Hanna et al. [52] (scheme 2a). In a typical experiment, a mixture of 3-methoxyphenethylamine (1.0 g, 6.6 mmol), hydrobromic acid (10 ml) and acetic acid (10 ml) was stirred and heated under reflux for 6 h. The cooled mixture was concentrated under reduced pressure to obtain a residue, which was used in subsequent reactions without further purification; crude yield, 97%.

## 3.3. General method for the synthesis of substituted N-benzylisatins (**8a–n**) (method B)

This was done using the method of Vine et al. [48] with some modifications (scheme 1a). The desired isatin (1 equiv) was taken up in anhydrous acetonitrile or DMF (1 ml per 0.1 mmol of isatin). Solid $K_2CO_3$ (1.2 equiv) was added in one portion, and the dark-coloured suspension was stirred at room temperature for 1 h. The appropriate benzyl halide (1.2 equiv) and KI (0.2 equiv) were added, and the reaction mixture was stirred at 80°C for 5–18 h, until the isatin starting material had been consumed as revealed by TLC. The reaction mixture was decanted into HCl (0.5 M, 50 ml) and extracted with methylene chloride (20 ml × 2), dried over anhydrous sodium sulfate and the solvent removed under reduced pressure. The crude product obtained was purified by flash chromatography using isocratic elution with hexane : ethyl acetate giving yellow to red crystals. Yields were between 80 and 95%.

## 3.4. Synthesis of 5,7-dibromoisatin (**7c**) (method C)

The synthesis of 5,7-dibromoisatin was based on the method of Kumar et al. [25] (scheme 1b). Isatin (9.0 g, 61.2 mmol, 1 equiv) was warmed in ethanol (95%, 100 ml) with stirring until it dissolved. Bromine (3.0 equiv, 16.3 g, 183.6 mmol, 9.4 ml) was added dropwise to the stirred isatin solution while maintaining the temperature of the reaction mixture between 70 and 75°C. The solution was cooled to room temperature and placed on ice for 30 min. The resulting precipitate was washed with water and cold ethanol and then recrystallized from ethanol to yield bright orange-red crystals of 5,7-dibromoisatin (66%), m.p. 248–250°C (lit. 248–250°C).

(a)

**7a**: R = H
**7b**: R = Cl

i

**8**

(b)

(R=H)

ii

**7c**

i: Arylalkyl halide, K$_2$CO$_3$, DMF, heat.
ii: Bromine, EtOH;

**8a**: R = R′ = H; Ar = 4-fluorophenyl
**8b**: R = R′ = H; Ar = 4-chlorophenyl
**8c**: R = R′ = H; Ar = 4-bromophenyl
**8d**: R = R′ = H; Ar = 3,4-dichlorophenyl
**8e**: R = R′ = H; Ar = 4-methylphenyl
**8f**: R = R′ = H; Ar = 2-nitrophenyl
**8g**: R = R′ = H; Ar = 2-naphthyl

**8h**: R = Cl; R′ = H; Ar = 4-fluorophenyl
**8i**: R = Cl; R′ = H; Ar = 4-chlorophenyl
**8j**: R = Cl; R′ = H; Ar = 4-bromophenyl
**8k**: R = Cl; R′ = H; Ar = 3,4-dichlorophenyl
**8l**: R = Cl; R′ = H; Ar = 4-methylphenyl
**8m**: R = Cl; R′ = H; Ar = 2-nitrophenyl
**8n**: R = Cl; R′ = H; Ar = 2-naphthyl

(c)

**9a**: R$_1$ = R$_2$ = OMe; R$_3$ = H
**9b**: R$_1$ = R$_2$ = R$_3$ = OMe

**10a, 10b**

**11a, 11b**

iii. Nitromethane, NH$_4$OAc, HOAc reflux; iv. Zn/HCl, MeOH, 0–5°C.

**Scheme 1.** Synthesis of major fragments. (a) Method B, (b) Method C and (c) Method F.

## 3.5. General method for the synthesis of 6′-hydroxy-3′,4′-dihydro-2′H-spiro[indoline-3,1′-isoquinolin]-2-ones (**1a–q**) and 8′-hydroxy-3′,4′-dihydro-2′H-spiro[indoline-3,1′-isoquinolin]-2-ones (**2a–q**) (method D)

The compounds were synthesized by the phenolic Pictet–Spengler reaction reported by Kametani et al. [47] and modified by Ngo Hanna et al. [52] (scheme 2b). In a typical experiment, crude 3-hydroxyphenethylamine (0.9 g, 6.6 mmol, 1 equiv), obtained as described in Method A above, and the appropriate isatin (1 equiv, 6.6 mmol) were dissolved in absolute ethanol (10 ml) and triethylamine (1 ml) was added. The reaction mixture was stirred and heated under reflux for 7–10 h. Upon completion, the reaction was concentrated under reduced pressure to remove the solvent. Distilled water was added to the viscous mass and the product which precipitated out was extracted into ethyl acetate (3 × 30 ml). The combined organic extracts were dried over anhydrous sodium sulfate and concentrated to minimum volume. The crude product was further purified by column chromatography using the appropriate solvent system. The final product obtained was recrystallized from absolute ethanol. In most cases, the reaction provided two regioisomers as earlier documented by Kametani et al. [47]: the 8′-hydroxy compound, which had a shorter retention time, and the 6′-hydroxy isomer. The combined yields ranged between 40 and 99%.

**Scheme 2.** Synthesis of target compounds. (*a*) Method A, (*b*) Method D, (*c*) Method E and (*d*) Method G.

## 3.6. 8′-Hydroxy-3′,4′-dihydro-2′H-spiro[indoline-3,1′-isoquinolin]-2-one (**1a**)

Method D. Prepared from 3-hydroxyphenethylamine (0.9 g, 6.61 mmol, 1 equiv) and isatin (0.96 g, 6.61 mmol, 1 equiv). The crude product was purified on flash chromatography (hexane : ethyl acetate—60 : 40) and recrystallized from methanol. Yield, 0.83 g, 45% (white solid). M.p. 256–258°C. $^{1}$H NMR (DMSO-$d_6$, 700 MHz): $\delta$ ppm 2.80 (t, $J$ = 5.8 Hz, 2H, H4′a, H4′b), 3.07 (dt, $J$ = 12.9, 6.3 Hz, 1H, H3′a), 3.13–3.16 (m, 1H, H3′b), 6.42 (dd, $J$ = 8.0, 1.2 Hz, 1H, H7′), 6.63 (dd, $J$ = 7.60, 1.1 Hz, 1H, H5′), 6.80 (ddd, $J$ = 7.3, 6.0, 1.3 Hz, 2H, H5, H7), 6.88 (m, 1H, H4), 6.98 (t, $J$ = 7.7 Hz, 1H, H6′), 7.13 (td, $J$ = 7.6, 1.3 Hz, 1H, H6), 9.12 (s, 1H, 8′-O<u>H</u>), 10.19 (s, 1H, H1). $^{13}$C NMR (DMSO-$d_6$, 175 MHz): $\delta$ ppm 28.8 (C4′), 38.4 (C3′), 61.9 (C3/C1′), 108.8 (C7), 112.3 (C7′), 119. 6 (C5′), 120.6 (C5), 122.5 (C8′a), 123.1 (C4), 127.2 (C6′), 127.6 (C6), 135.6 (C3a), 138.4 (C4′a), 142.8 (C7a), 153.9 (C8′), 179.3 (C2). **FTMS + cESI:** $m/z$ 267.11 [M + 1]$^{+}$.

## 3.7. 5-Chloro-6′-hydroxy-3′,4′-dihydro-2′H-spiro[indoline-3,1′-isoquinolin]-2-one (**2b**)

Method D. Prepared from 3-hydroxyphenethylamine (1.2 g, 6.61 mmol) and 5-chloroisatin (0.9 g, 6.61 mmol). The crude product was purified by chromatography on a short column (hexane : ethyl

acetate—60 : 40) and recrystallized from methanol. Yield, 1.0 g, 48% (white solid). M.p. 253–255°C. **[1]H NMR** (DMSO-d$_6$, 700 MHz): δ ppm 2.65 (dt, $J$ = 16.0, 3.4 Hz, 1H, H4′a), 2.86 (ddd, $J$ = 15.6, 9.3, 5.4 Hz, 1H, H4′b), 2.97 (dt, $J$ = 12.1, 4.7 Hz, 1H, H3′a), 3.05 (br s, 1H, H2′), 3.60 (ddd, $J$ = 12.8, 9.6, 4.1 Hz, 1H, H3′b), 6.24 (d, $J$ = 8.4 Hz, 1H, H8′), 6.42 (dd, $J$ = 8.4, 2.6 Hz, 1H, H7′), 6.55 (d, $J$ = 2.6 Hz, 1H, H5′), 6.89 (d, $J$ = 8.3 Hz, 1H, H7), 7.02 (d, $J$ = 2.2 Hz, 1H, H4), 7.26 (dd, $J$ = 8.3, 2.2 Hz, 1H, H6), 9.29 (s, 1H, 6′-O<u>H</u>), 10.37 (s, 1H, H1). **[13]C NMR** (DMSO-d$_6$, 175 MHz): δ ppm 28.8 (C4′), 38.0 (C3′), 63.3 (C3/C1′), 110.9 (C7), 113.7 (C7′), 115. 3 (C5′), 124.5 (C8′a), 125.0 (C4), 125.7 (C5), 126.9 (C8′), 128.3 (C6), 137.6 (C4′a), 138.1 (C3a), 141.1 (C7a), 156.0 (C6′), 180.0 (C2). **FTMS + cESI:** $m/z$ 301.07 [M + 1]$^+$

## 3.8. 5,7-Dibromo-8′-hydroxy-3′,4′-dihydro-2′H-spiro[indoline-3,1′-isoquinolin]-2-one (**1c**)

Method D. Prepared from **7c** (1.5 g, 5 mmol, 1 equiv) and 3-hydroxyphenethylamine (0.81 g, 6 mmol). The product was separated from its 6′-OH regioisomer by column chromatography (hexane : ethyl acetate—80 : 20). Yield, 0.33 g, 16% (brown solid). M.p. 235–238°C. **[1]H NMR** (CD$_3$OD 600 MHz): δ ppm 2.95–2.92 (m, 2H, H4′a, H4′b), 3.16–3.22 (m, 1H, H3′a), 3.31 (d, $J$ = 5.2 Hz, 1H, H3′b), 6.53–6.55 (m, 1H, H7′), 6.74 (d, $J$ = 7.7 Hz, 1H, H5′), 7.06–7.11 (m, 2H, H4, H6′), 7.56 (d, $J$ = 1.8, Hz, 1H, H6). **[13]C NMR** (DMSO-d$_6$, 175 MHz): δ ppm 28.1 (C4′), 38.5 (C3′), 63.5 (C3/C1′), 102.5 (C7), 112.3 (C7′), 113.9 (C5), 120 (C5′), 120.2 (C8′a), 125.3 (C4), 128.2 (C6′), 132.8 (C6), 137.9 (C3a), 138.0 (C4′a), 141.4 (C7a), 154.1 (C8′), 179.9 (C2). **FTMS + cESI:** $m/z$ 426.93 [M + 1]$^+$.

## 3.9. 1-(4-Fluorobenzyl)-8′-hydroxy-3′,4′-dihydro-2′H-spiro[indoline-3,1′-isoquinolin]-2-one (**1d**) and 1-(4-fluorobenzyl)-6′-hydroxy-3′,4′-dihydro-2′H-spiro[indoline-3,1′-isoquinolin]-2-one (**2d**)

Method D. Prepared from 1-(4-fluorobenzyl)indoline-2,3-dione, **8a** (0.4 g, 1.6 mmol, 1 equiv), and 3-hydroxyphenethylamine (0.2 g, 1.6 mmol). The reaction afforded compounds **1d** and **2d** that were separated by column chromatography (hexane : ethyl acetate—80 : 20).

**1d.** Yield, 0.2 g, 33% (brown solid). M.p. 119–121°C. **[1]H NMR** (CD$_3$OD, 600 MHz): δ ppm 2.93–3.06 (m, 2H, H4′a, H4′b), 3.32–3.37 (m, 2H, H3′a, H3′b), 4.84 (d, $J$ = 15.9 Hz, 1H, –C<u>H$_2$</u>–Ar), 5.16 (d, $J$ = 15.9 Hz, 1H, C<u>H$_2$</u>–Ar), 6.54 (dd, $J$ = 8.1, 1.1 Hz, 1H, H7′), 6.75–6.80 (m, 2H, H7, H5′), 6.97 (td, $J$ = 7.6, 1.0 Hz, 1H, H5), 7.07 (ddt, $J$ = 7.8, 4.0, 2.3 Hz, 4H, H4, H6′, H3″, H5″), 7.18 (td, $J$ = 7.7, 1.3 Hz, 1H, H6), 7.49–7.53 (m, 2H, H2″, H6″). **[13]C NMR** (CD$_3$OD, 150 MHz): δ ppm 28.3 (C4′), 38.7 (C3′), 42.8 (–C<u>H$_2$</u>–Ar), 62.3 (C3/C1′), 108.9 (C7), 112.3 (C7′), 114.9 (2C, C3″, C5″), 120.8 (C5′), 120.8 (C8′a), 122.2 (C5), 123.2 (C4), 127.9 (2C, C6, C6′), 129.0 (2C, C2″, C6″), 132.1 (C1″), 133.7 (C3a), 138.0 (C4′a), 143.1 (C7a), 154.0 (C8′), 163.2 (C4″), 179.0 (C2). **FTMS + cESI:** $m/z$ 375.15 [M + 1]$^+$.

**2d.** Yield, 0.2 g, 33% (brown solid). M.p. 220–222°C. **[1]H NMR** (CD$_3$OD, 600 MHz): δ ppm 2.88 (dt, $J$ = 16.4, 4.5, 1H, H4′a), 3.04 (ddd, $J$ = 15.2, 9.1, 5.5, 1H, H4′b), 3.18–3.23 (m, 1H, H3′a), 3.83–3.88 (m, 1H, H3′b), 4.90 (d, $J$ = 15.6 Hz, 1H, –C<u>H$_2$</u>–Ar), 5.01 (d, $J$ = 15.6 Hz, 1H, C<u>H$_2$</u>–Ar), 6.22 (d, $J$ = 8.5 Hz, 1H, H8′), 6.45 (dd, $J$ = 8.5, 2.6 Hz, 1H, H7′), 6.65 (d, $J$ = 2.6, 1H, H5′), 6.97 (d, $J$ = 7.9, 1H, H7), 7.06 (td, $J$ = 7.6, 1.0 Hz, 1H, H5), 7.07–7.11 (m, 2H, H2″, H6″), 7.19 (dd, $J$ = 7.5, 1.3 Hz, 1H, H4), 7.28 (td, $J$ = 7.8, 1.3 Hz, 1H, H6), 7.42–7.47 (m, 2H, H3″, H5″). **[13]C NMR** (CD$_3$OD, 150 MHz): δ ppm 28.3 (C4′), 38.3 (C3′), 42.4 (–C<u>H$_2$</u>–Ar), 63.3 (C3/C1′), 109.2 (C7), 113.6 (C7′), 115.1 (2C, C3″, C5″), 115.2 (C5′), 123.2 (C5), 124.3 (C4), 124.9 (C8′a), 127.0 (C8′), 128.8 (C6), 129.2 (2C, C2″, C6″), 132.2 (C1″), 134.6 (C3a), 137.4 (C4′a), 142.5 (C7a), 156.4 (C6′), 163.2 (C4″), 179.2 (C2). **FTMS + cESI:** $m/z$ 375.15 [M + 1]$^+$.

## 3.10. 1-(4-Chlorobenzyl)-8′-hydroxy-3′,4′-dihydro-2′H-spiro[indoline-3,1′-isoquinolin]-2-one (**1e**) and 1-(4-chlorobenzyl)-6′-hydroxy-3′,4′-dihydro-2′H-spiro[indoline-3,1′-isoquinolin]-2-one (**2e**)

Method D. Prepared from 1-(4-chlorobenzyl)indoline-2,3-dione, **8b** (0.8 g, 3.0 mmol), and 3-hydroxyphenethylamine (0.6 g, 4.4 mmol). The regioisomeric products **1e** and **2e** were separated by column chromatography (hexane : ethyl acetate—80 : 20).

**1e.** Yield, 0.4 g, 24% (brown solid). M.p. 95–98°C. **[1]H NMR** (CD$_3$OD, 600 MHz): δ ppm 2.84 (dt, $J$ = 16.7, 4.9 Hz, 1H, H4′a), 2.90 (dt, $J$ = 16.7, 4.9 Hz, 1H, H4′b), 3.22 (m, 2H, H3′a, H3′b), 4.71 (d, $J$ = 16.1 Hz, 1H, –C<u>H$_2$</u>–Ar), 5.04 (d, $J$ = 16.1 Hz, 1H, C<u>H$_2$</u>–Ar), 6.42 (dd, $J$ = 8.0, 1.1 Hz, 1H, H7′), 6.63–6.66 (m, 2H, H7,

H5′), 6.85 (td, *J* = 7.5, 1.0 Hz, 1H, H5), 6.93–6.97 (m, 2H, H4, H6), 7.06 (td, *J* = 7.8, 1.3 Hz, 1H, H6′), 7.22–7.24 (m, 2H, H3″, H5″), 7.36 (d, *J* = 8.4, 2H, H2″, H6″). <sup>13</sup>C NMR (CD₃OD, 150 MHz): δ ppm 28.3 (C4′), 38.7 (C3′), 42.8 (–CH₂–Ar), 62.3 (C3/C1′), 109.0 (C7), 120.0 (2C, C5′, C7′), 120.7 (C8′a), 122.3 (C4), 122.3 (C5), 128.0 (C6′), 128.1 (C6), 128.3 (2C, C3″, C5″), 128.7 (2C, C2″, C6″), 132.8 (C4″), 133.6 (C3a), 135.0 (C1″), 137.9 (C4′a), 143.0 (C7a), 154.0 (C8′), 178.8 (C2). **FTMS + cESI:** *m/z* 391.12 [M + 1]⁺.

**2e.** Yield, 0.8 g, 47% (white solid). M.p. 218–220°C. **¹H NMR** (CD₃OD, 600 MHz): δ ppm 2.88 (dt, *J* = 16.4, 4.5, 1H, H4′a), 3.04 (ddd, *J* = 15.3, 9.2, 5.5 Hz 1H, H4′b), 3.18–3.24 (m, 1H, H3′a), 3.85 (ddd, *J* = 12.7, 9.1, 4.6 Hz 1H, H3′b), 4.92 (d, *J* = 15.8 Hz, 1H, –CH₂–Ar), 5.00 (d, *J* = 15.8 Hz, 1H, CH₂–Ar), 6.23 (d, *J* = 8.5 Hz, 1H, H8′), 6.46 (dd, *J* = 8.5, 2.6 Hz, 1H, H7′), 6.66 (d, *J* = 2.6, 1H, H5′), 6.94 (dd, *J* = 7.9, 1H, H7), 7.06 (td, *J* = 7.5, 1.0 Hz, 1H, H5), 7.19 (dd, *J* = 7.5, 1.2 Hz, 1H, H4), 7.28 (td, *J* = 7.7, 1.3 Hz, 1H, H6), 7.33–7.36 (m, 2H, H2″, H6″), 7.40 (m, 2H, H3″, H5″). **¹³C NMR** (CD₃OD, 150 MHz): δ ppm 28.3 (C4′), 38.3 (C3′), 42.4 (–CH₂–Ar), 63.3 (C3/C1′), 109.2 (C7), 113.6 (C7′), 115.2 (C5′), 123.3 (C5), 124.3 (C4), 124.8 (C8′a), 127.0 (C8′), 128.5 (2C, C2″, C6″), 128.8 (C6), 129.1 (2C, C3″, C5″), 133.2 (C4″), 134.6 (C3a), 134.9 (C1″), 137.4 (C4′a), 142.5 (C7a), 156.4 (C6′), 179.2 (C2). **FTMS + cESI:** *m/z* 391.12 [M + 1]⁺.

## 3.11. Synthesis of 1-(4-bromobenzyl)-8′-hydroxy-3′,4′-dihydro-2′H-spiro[indoline-3,1′-isoquinolin]-2-one (**1f**) and 1-(4-bromobenzyl)-6′-hydroxy-3′,4′-dihydro-2′H-spiro[indoline-3,1′-isoquinolin]-2-one (**2f**)

Method D. Prepared from 3-hydroxyphenethylamine (1.0 g, 7.1 mmol) and 1-(4-bromobenzyl)indoline-2,3-dione, **8c** (1.5 g, 4.7 mmol). The regioisomers **1f** and **2f** were separated by column chromatography (hexane : ethyl acetate—80 : 20).

**1f.** Yield, 0.3 g, 15% (brown solid). M.p. 81–83°C. **¹H NMR** (CD₃OD, 600 MHz): δ ppm 2.97–3.01 (m, 2H, H4′a, H4′b), 3.34 (m, 2H, H3′a, H3′b), 4.81 (d, *J* = 16.3 Hz, 1H, –CH₂–Ar), 5.14 (d, *J* = 16.3 Hz, 1H, CH₂–Ar), 6.54 (dd, *J* = 8.0, 1.1 Hz, 1H, H7′), 6.75–6.78 (m, 2H, H7, H5′), 6.97 (td, *J* = 7.5, 1.0 Hz, 1H, H5), 7.07–7.09 (m, 1H, H4), 7.19 (td, *J* = 7.8, 1.3 Hz, 1H, H6′), 7.29–7.31 (m, H6), 7.41–7.44 (m, 2H, H3″, H5″), 7.50 (m, 2H, H2″, H6″). **¹³C NMR** (CD₃OD, 150 MHz): δ ppm 28.2 (C4′), 38.7 (C3′), 42.8 (–CH₂–Ar), 62.3 (C3/C1′), 109.0 (C7), 112.3 (C7′), 114.6 (C4″), 120.0 (C5′), 120.7 (C8′a), 122.3 (C5), 123.3 (C4), 128.1 (C6′), 128.7 (C6), 129.0 (2C, C3″, C5″), 131.3 (2C, C2″, C6″), 133.9 (C1″), 135.5 (C3a), 137.9 (C4′a), 143.0 (C7a), 154.0 (C8′), 178.8 (C2). **FTMS + cESI:** *m/z* 435.07 [M + 1]⁺

**2f.** Yield, 0.82 g, 40% (white solid). M.p. 207–210°C. **¹H NMR** (CD₃OD, 600 MHz): δ ppm 2.88 (dt, *J* = 16.4, 4.5, 1H, H4′a), 3.04(ddd, *J* = 15.2, 9.2, 5.5 Hz 1H, H4′b), 3.17–3.24 (m, 1H, H3′a), 3.82–3.89 (m, 1H, H3′b), 4.90 (d, *J* = 15.8 Hz, 1H, –CH₂–Ar), 4.98 (d, *J* = 15.8 Hz, 1H, CH₂–Ar), 6.24 (d, *J* = 8.5 Hz, 1H, H8′), 6.46 (dd, *J* = 8.5, 2.6 Hz, 1H, H7′), 6.66 (d, *J* = 2.6 Hz, 1H, H5′), 6.94 (dd, *J* = 7.8 Hz, 1H, H7), 7.06 (td, *J* = 7.5, 1.0 Hz, 1H, H5), 7.19 (dd, *J* = 7.5, 1.3 Hz, 1H, H4), 7.28 (td, *J* = 7.8, 1.3 Hz, 1H, H6), 7.33–7.36 (m, 2H, H2″, H6″), 7.50–7.54 (m, 2H, H3″, H5″).**¹³C NMR** (CD₃OD, 150 MHz): δ ppm 28.3 (C4′), 38.3 (C3′), 42.4 (–CH₂–Ar), 63.3 (C3/C1′), 109.2 (C7), 113.6 (C7′), 115.2 (C5′), 121.1 (C4″), 123.3 (C5), 124.3 (C4), 124.8 (C8′a), 127.1 (C8′), 128.8 (C6), 129.1 (2C, C3″, C5″), 131.5 (2C, C2″, C6″), 134.6 (C3a), 134.6 (C1″), 137.4 (C4′a), 142.5 (C7a), 156.4 (C6′), 179.2 (C2). **FTMS + cESI:** *m/z* 435.07 [M + 1]⁺.

## 3.12. 1-(3,4-Dichlorobenzyl)-8′-hydroxy-3′,4′-dihydro-2′H-spiro[indoline-3,1′-isoquinolin]-2-one (**1g**) and 1-(3,4-dichlorobenzyl)-6′-hydroxy-3′,4′-dihydro-2′H-spiro[indoline-3,1′-isoquinolin]-2-one (**2g**)

Method D. Prepared from 3-hydroxyphenethylamine (0.82 g, 6.0 mmol) and 1-(3,4-dichlorobenzyl)indoline-2,3-dione, **8d** (1.22 g, 4.0 mmol). The reaction afforded compounds **1g** and **2g** that were separated by column chromatography (hexane : ethyl acetate—80 : 20).

**1g.** Yield, 0.4 g, 24% (brown solid). M.p. 109–113°C. **¹H NMR** (CD₃OD, 600 MHz): δ ppm 2.93–3.06 (m, 2H, H4′a, H4′b), 3.27–3.37 (m, 2H, H3′a, H3′b), 4.85 (d, *J* = 16.3 Hz, 1H, –CH₂–Ar), 5.12 (d, *J* = 16.3 Hz, 1H, CH₂–Ar), 6.54 (dd, *J* = 8.0, 1.1 Hz, 1H, H7′), 6.76 (dd, *J* = 7.6, 1.2 Hz, 1H, H5′), 6.81 (d, *J* = 7.8 Hz, 1H, H7), 6.99 (td, *J* = 7.5, 1.0 Hz, 1H, H5), 7.05–7.10 (m, 2H, H4, H6′), 7.22 (td, *J* = 7.7, 1.3 Hz, 1H, H6), 7.42 (dd, *J* = 8.3, 2.1 Hz, 1H, H6″), 7.49 (d, *J* = 8.3, 1H, H5″), 7.70 (d, *J* = 2.1 Hz, 1H, H2″). **¹³C NMR** (CD₃OD, 150 MHz): δ ppm 28.3 (C4′), 38.7 (C3′), 42.4 (–CH₂–Ar), 62.2 (C3/C1′), 108.8 (C7), 114.4 (C7′), 120.0 (C5′), 120.8 (C8′a), 122.4 (C5), 123.3 (C4), 127.0 (C6″), 127.9 (C6′), 128.1 (C6), 129.3 (C2″), 130.3 (C5″), 130.9 (C3″), 132.1 (C4″), 133.8 (C3a), 137.2 (C1″), 138.0 (C4′a), 142.9 (C7a), 154.0 (C8′), 179.0 (C2). **FTMS + cESI:** *m/z* 425.08 [M + 1]⁺.

**2g.** Yield, 0.9 g, 53% (white solid). M.p. 197–199°C. $^1$**H NMR** (CD$_3$OD, 600 MHz): $\delta$ ppm 2.88 (dt, $J$ = 16.4, 4.4 Hz, 1H, H4'a), 3.00–3.08 (m, 1H, H4'b), 3.20 (ddd, $J$ = 12.8, 5.5, 4.3 Hz 1H, H3'a), 3.86 (ddd, $J$ = 12.8, 9.3, 4.6 Hz 1H, H3'b), 4.90 (d, $J$ = 16.0 Hz, 1H, –C$\underline{H_2}$–Ar), 5.00 (d, $J$ = 16.0 Hz, 1H, C$\underline{H_2}$–Ar), 6.23 (d, $J$ = 8.5 Hz, 1H, H8'), 6.47 (dd, $J$ = 8.5, 2.6 Hz, 1H, H7'), 6.66 (d, $J$ = 2.6 Hz, 1H, H5'), 6.97 (d, $J$ = 7.9 Hz, 1H, H7), 7.08 (td, $J$ = 7.5, 1.0 Hz, 1H, H5), 7.21 (dd, $J$ = 7.5, 1.3 Hz, 1H, H4), 7.28 (td, $J$ = 7.8, 1.3 Hz, 1H, H6), 7.35 (dd, $J$ = 8.3, 2.1 Hz, 1H, H6''), 7.52 (d, $J$ = 8.3 Hz, 1H, H5''), 7.59 (d, $J$ = 2.1 Hz, 1H, H2'').$^{13}$**C NMR** (CD$_3$OD, 150 MHz): $\delta$ ppm 28.3 (C4'), 38.3 (C3'), 41.9 (–C$\underline{H}_2$–Ar), 63.2 (C3/C1'), 109.0 (C7), 113.7 (C7'), 115.2 (C5'), 123.4 (C5), 124.4 (C8'a), 124.8 (C4), 127 (C8'), 127.1 (C6''), 128.9 (C6), 129.2 (C2''), 130.5 (C5''), 131.2 (C4''), 132.3 (C1''), 134.5 (C3a), 137.1 (C3''), 137.4 (C4'a), 142.3 (C7a), 156.4 (C6'), 179.2 (C2). **FTMS + cESI:** $m/z$ 425.08 [M + 1]$^+$.

## 3.13. 8'-Hydroxy-1-(4-methylbenzyl)-3',4'-dihydro-2'H-spiro[indoline-3,1'-isoquinolin]-2-one (**1h**) and 6'-hydroxy-1-(4-methylbenzyl)-3',4'-dihydro-2'H-spiro[indoline-3,1'-isoquinolin]-2-one (**2h**)

Method D. Prepared from 1-(4-methylbenzyl)indoline-2,3-dione, **8e** (053 g, 2.0 mmol), and 3-hydroxyphenethylamine (0.4 g, 2.8 mmol). The products **1h** and **2h** were separated by column chromatography (hexane : ethyl acetate—80 : 20).

**1h.** Yield, 0.1 g, 12% (brown solid). M.p. 110–112°C. $^1$**H NMR** (CD$_3$OD, 600 MHz): $\delta$ ppm 2.33 (s, Ar–C$\underline{H_3}$), 2.96 (dt, $J$ = 16.4, 4.7 Hz, 1H, H4'a), 3.03 (dt, $J$ = 16.4, 4.9 Hz, 1H, H4'b), 3.34 (m, 2H, H3'a, H3'b), 4.83 (d, $J$ = 15.8 Hz, 1H, –C$\underline{H_2}$–Ar), 5.1 (d, $J$ = 15.8 Hz, 1H, C$\underline{H_2}$–Ar), 6.54 (dd, $J$ = 8.0, 1.1 Hz, 1H, H7'), 6.74–6.78 (m, 2H, H7, H5'), 6.95 (td, $J$ = 7.5, 1.0 Hz, 1H, H5), 7.05–7.09 (m, 2H, H4, H6'), 7.16–7.18 (m, 3H, H6, H3'', H5''), 7.36–7.38 (m, 2H, H2'', H6''). $^{13}$**C NMR** (CD$_3$OD, 150 MHz): $\delta$ ppm 19.8 (Ar–C$\underline{H}_3$), 28.3 (C4'), 38.7 (C3'), 43.4 (–C$\underline{H}_2$–Ar), 62.3 (C3/C1'), 109.2 (C7), 112.3 (C7'), 119.9 (2C, C5'), 120.9 (C8'a), 122.1 (C5), 123.1 (C4), 127.0 (2C, C2'', C6''), 127.9 (2C, C6,C6'), 128.8 (2C, C3'', C5''), 1330 (C1''), 133.7 (C3a), 136.7 (C4''), 137.9 (C4'a), 143.3 (C7a), 154.1 (C8'), 179.0 (C2). **FTMS + cESI:** $m/z$ 371.17 [M + 1]$^+$.

**2h.** Yield, 0.5 g, 62% (white solid). M.p. 190–193°C. $^1$**H NMR** (CD$_3$OD, 600 MHz): $\delta$ ppm 2.33 (s, 3H, Ar–C$\underline{H_3}$), 2.88 (dt, $J$ = 16.4, 4.6 Hz, 1H, H4'a), 3.03 (ddd, $J$ = 16.4, 9.0, 5.5 Hz 1H, H4'b), 3.20 (dt, $J$ = 12.8, 5.1 Hz, 1H, H3'a), 3.84 (ddd, $J$ = 12.8, 9.1, 4.6 Hz, 1H, H3'b), 4.82 (d, $J$ = 15.6 Hz, 1H, –C$\underline{H_2}$–Ar), 5.01 (d, $J$ = 15.6 Hz, 1H, C$\underline{H_2}$–Ar), 6.24 (d, $J$ = 8.5 Hz, 1H, H8'), 6.45 (dd, $J$ = 8.5, 2.6 Hz, 1H, H7'), 6.65 (d, $J$ = 2.6 Hz, 1H, H5'), 6.93 (dd, $J$ = 7.9 Hz, 1H, H7), 7.03 (td, $J$ = 7.5, 1.0 Hz, 1H, H5), 7.17 (m, 3H, H4, H3'', H5''), 7.25 (td, $J$ = 7.8, 1.3 Hz, 1H, H6), 7.28 (d, 2H, $J$ = 8.0, H2'', H6''). $^{13}$**C NMR** (CD$_3$OD, 150 MHz): $\delta$ ppm 28.3 (C4'), 38.3 (C3'), 42.9 (–C$\underline{H}_2$–Ar), 63.3 (C3/C1'), 109.4 (C7), 113.6 (C7'), 115.1 (C5'), 123.1 (C5), 124.2 (C4), 124.9 (C8'a), 127.1 (3C, C8', C2'', C6''), 128.7 (C6), 129.0 (2C, C3'', C5''), 133.0 (C1''), 134.6 (C3a), 137.2 (C4''), 137.3 (C4'a), 142.7 (C7a), 156.3 (C6'), 179.2 (C2). **FTMS + cESI:** $m/z$ 371.17 [M + 1]$^+$.

## 3.14. 8'-Hydroxy-1-(2-nitrobenzyl)-3',4'-dihydro-2'H-spiro[indoline-3,1'-isoquinolin]-2-one (**1i**) and 6'-hydroxy-1-(2-nitrobenzyl)-3',4'-dihydro-2'H-spiro[indoline-3,1'-isoquinolin]-2-one (**2i**)

Method D. Prepared from 1-(2-nitrobenzyl)indoline-2,3-dione, **8f** (0.8 g, 3.0 mmol), and 3-hydroxyphenethylamine (0.6 g, 4.3 mmol). The reaction afforded two regioisomers, **1i** and **2i**, that were separated by column chromatography (hexane : ethyl acetate—80 : 20).

**1i.** Yield, 0.1 g, 8% (brown solid). M.p. 122–124°C. $^1$**H NMR** (CD$_3$OD, 600 MHz): $\delta$ ppm 2.94–3.04 (m, 2H, H4'a, H4'b), 3.36 (m, 2H, H3'a, H3'b), 5.27 (d, $J$ = 18.0 Hz, 1H, –C$\underline{H_2}$–Ar), 5.49 (d, $J$ = 18.0 Hz, 1H, C$\underline{H_2}$–Ar), 6.57 (dd, $J$ = 8.0, 1.1 Hz, 1H, H7'), 6.70 (d, $J$ = 7.9 Hz, 1H, H7), 6.78 (dd, $J$ = 7.6 Hz, 1H, H5'), 7.02 (td, $J$ = 7.5, 1.0 Hz, 1H, H5), 7.09 (td, $J$ = 7.8 Hz, 1H, H6'), 7.13 (dd, $J$ = 7.4, 1.3 Hz, 1H, H4), 7.21 (td, $J$ = 7.8, 1.3 Hz, 1H, H6), 7.55 (td, $J$ = 7.7, 1.4 Hz, 1H, H6'), 7.65 (td, $J$ = 7.6, 1.3 Hz, 1H, H6''), 8.22 (dd, $J$ = 8.2, 1.3 Hz, 1H, H3''). $^{13}$**C NMR** (CD$_3$OD, 150 MHz): $\delta$ ppm 28.3 (C4'), 38.6 (C3'), 41.5 (–C$\underline{H}_2$–Ar), 62.3 (C3/C1'), 108.7 (C7), 112.3 (C7'), 120.1 (C5'), 120.7 (C8'a), 122.6 (C5), 123.4 (C4), 124.9 (C3''), 128.0 (C6'), 128.1 (C4''), 128.2 (2C, C6, C6''), 131.8 (C1''), 133.6 (C3a), 133.7 (C5''), 138.1 (C4'a), 142.9 (C7a), 148.0 (C2''), 153.9 (C8'), 179.3 (C2). **FTMS + cESI:** $m/z$ 402.14 [M + 1]$^+$.

**2i.** Yield, 0.6 g, 50% (reddish brown solid). M.p. 115–118°C. $^1$**H NMR** (CD$_3$OD, 600 MHz): $\delta$ ppm 2.88 (dt, $J$ = 16.5, 4.4, 1H, H4'a), 3.06 (ddd, $J$ = 15.3, 9.3, 5.6 Hz 1H, H4'b), 3.22 (ddd, $J$ = 12.8, 9.3, 4.6 Hz, 1H,

H3′a), 3.87 (ddd, $J$ = 12.8, 9.3, 4.6 Hz 1H, H3′b), 5.30–5.39 (m 2H, –CH$_2$–Ar), 6.39 (d, $J$ = 8.5 Hz, 1H, H8′), 6.51 (dd, $J$ = 8.5, 2.6 Hz, 1H, H7′), 6.66 (d, $J$ = 2.6, 1H, H5′), 6.83 (dd, $J$ = 7.9, 1H, H7), 7.10 (td, $J$ = 7.5, 1.0 Hz, 1H, H5), 7.25 (dd, $J$ = 7.5, 1.2 Hz, 1H, H4), 7.28 (td, $J$ = 7.8, 1.3 Hz, 1H, H6), 7.55 (td, $J$ = 7.8, 1.4 Hz, 1H, H4″), 7.65 (td, $J$ = 7.6, 1.4 Hz, 1H, H5″), 8.19 (dd, $J$ = 8.20, 1.4 Hz, 1H, H3″). $^{13}$C NMR (CD$_3$OD, 150 MHz): $\delta$ ppm 28.2 (C4′), 38.3 (C3′), 40.8 (–CH$_2$–Ar), 63.4 (C3/C1′), 109.0 (C7), 113.7 (C7′), 115.2 (C5′), 123.3 (C5), 124.5 (C4), 124.7 (C8′a), 124.5 (C3″), 127.2 (C8′), 127.6 (C6″), 128.3 (C4″), 129.1 (C6), 131.2 (C1″), 134.4 (C3a), 133.7 (C5″), 137.4 (C4′a), 142.5 (C7a), 148.3 (C2″), 156.5 (C6′), 179.4 (C2). **FTMS + cESI:** $m/z$ 402.14 [M + 1]$^+$.

## 3.15. 8′-Hydroxy-1-(naphthalen-2-ylmethyl)-3′,4′-dihydro-2′H-spiro[indoline-3,1′-isoquinolin]-2-one (**1j**) and 6′-Hydroxy-1-(naphthalen-2-ylmethyl)-3′,4′-dihydro-2′H-spiro[indoline-3,1′-isoquinolin]-2-one (**2j**)

Method D. Prepared from 1-(naphthalen-2-ylmethyl)indoline-2,3-dione, **8 g** (2.7 g, 9.4 mmol), and 3-hydroxyphenethylamine (1.54 g, 11.3 mmol). The reaction afforded compounds, **1j** and **2j**, that were separated by column chromatography (hexane : ethyl acetate—80 : 20).

**1j.** Yield, 0.1 g, 16% (brown solid). M.p. 127–129°C. $^1$H NMR (CD$_3$OD, 600 MHz): $\delta$ ppm 2.97 (dt, $J$ = 16.6, 4.8 Hz, 1H, H4′a), 3.0–3.07 (m, 1H, H4′b), 3.36 (m, 2H, H3′a, H3′b), 5.05–5.08 (m, 1H, –CH$_2$–Ar), 5.29 (dd, $J$ = 16.0 Hz, 1H, CH$_2$–Ar), 6.58 (dd, $J$ = 8.0, 1.1 Hz, 1H, H7′), 6.77 (dd, $J$ = 7.6, 1.1 Hz, 1H, H5′), 6.82 (dd, $J$ = 7.9, 1.1 Hz, 1H, H7), 6.95 (td, $J$ = 7.5, 1.0 Hz, 1H, H5), 7.08–7.11 (m, 2H, H4, H6′), 7.13–7.16 (m, 1H, H6), 7.47 (td, $J$ = 7.4, 6.7, 3.4 Hz, 2H, H6″, H7″), 7.60 (dd, $J$ = 8.5 Hz, 1H, H3″), 7.83–7.90 (m, 3H, H4″, H5″, H8″), 7.97 (s, 1H, H2″). $^{13}$C NMR (CD$_3$OD, 150 MHz): $\delta$ ppm 28.2 (C4′), 38.7 (C3′), 43.8 (–CH$_2$–Ar), 62.4 (C3/C1′), 109.4 (C7), 112.2 (C7′), 120.0 (C5′), 122.2 (C8′a), 122.6 (C5), 123.3 (C4), 125.1 (C3″), 125.5 (C6″), 125.6 (C1″), 125.8 (C7″), 127.2 (C8″), 127.5 (C5″), 128.0 (C4″), 128.1 (C6′), 128.1 (C8″a), 129.3 (C6), 132.9 (C4″a), 133.5 (C2″), 133.6 (C3a), 137.8 (C4′a), 143.4 (C7a), 154.1 (C8′), 179.0 (C2). **FTMS + cESI:** $m/z$ 407.18 [M + 1]$^+$.

**2j.** Yield, 1.5 g, 36% (brown solid). M.p. 148–150°C. $^1$H NMR (CD$_3$OD, 600 MHz): $\delta$ ppm 2.90 (dt, $J$ = 16.4, 4.6 Hz, 1H, H4′a), 3.05 (ddd, $J$ = 15.1, 9.0, 5.5 Hz, 1H, H4′b), 3.23 (dt, $J$ = 12.8, 5.0 Hz, 1H, H3′a), 3.89 (ddd, $J$ = 13.2, 9.0, 4.6 Hz, 1H, H3′b), 5.05 (d, $J$ = 15.7 Hz, 1H, –CH$_2$–Ar), 5.21 (d, $J$ = 15.7 Hz, 1H, CH$_2$–Ar), 6.30 (d, $J$ = 8.5 Hz, 1H, H8′), 6.47 (dd, $J$ = 8.5, 2.6 Hz, 1H, H7′), 6.67 (d, $J$ = 2.6, 1H, H5′), 6.99 (d, $J$ = 7.9, 1H, H7), 7.20 (dd, $J$ = 7.4, 1.2, 1H, H4), 7.23 (td, $J$ = 7.8, 1.2 Hz, 1H, H6), 7.45–7.52 (m, 3H, H3″, H6″, H7″), 7.82–7.87 (m, 3H, H4″, H5″, H8″), 7.89 (d, $J$ = 1.8 Hz, 1H, H1″). $^{13}$C NMR (CD$_3$OD, 150 MHz): $\delta$ ppm 28.3 (C4′), 38.4 (C3′), 43.3 (–CH$_2$–Ar), 63.4 (C3/C1′), 109.4 (C7), 115.2 (C5′), 123.2 (C5), 124.3 (C4), 124.9 (C8′a), 125.0 (C7″), 125.7 (C6″), 125.9 (C1″), 126.0 (C3″), 127.1 (C8′), 127.3 (C8″), 127.4 (C5″), 128.1 (C8″a), 128.8 (C6), 133.0 (C3a), 133.5 (C2″), 134.6 (C4″a), 137.4 (C4′a), 142.7 (C7a), 156.4 (C6′), 179.3 (C2). **FTMS + cESI:** $m/z$ 407.18 [M + 1]$^+$.

## 3.16. 5-Chloro-1-(4-fluorobenzyl)-8′-hydroxy-3′,4′-dihydro-2′H-spiro[indoline-3,1′-isoquinolin]-2-one (**1k**) and 5-chloro-1-(4-fluorobenzyl)-6′-hydroxy-3′,4′-dihydro-2′H-spiro[indoline-3,1′-isoquinolin]-2-one (**2k**)

Method D. Prepared from 5-chloro-1-(4-fluorobenzyl)indoline-2,3-dione, **8h** (1.46 g, 5.0 mmol, 1 equiv), and 3-hydroxyphenethylamine (0.7 g, 5.0 mmol). The reaction products **1k** and **2k** were separated by column chromatography (hexane : ethyl acetate—80 : 20).

**1k.** Yield, 0.7 g, 35% (brown solid). M.p. 120–123°C. $^1$H NMR (CD$_3$OD, 600 MHz): $\delta$ ppm 2.97 (m, 2H, H4′a, H4′b), 3.21–3.28 (m, 1H, H3′a), 3.36 (dd, $J$ = 13.0, 5.3 Hz, 1H, H3′b), 4.81 (d, $J$ = 15.9 Hz, 1H, –CH$_2$–Ar), 5.14 (d, $J$ = 15.9 Hz, 1H, CH$_2$–Ar), 6.56 (dd, $J$ = 8.1, 1.1 Hz, 1H, H7′), 6.75 (d, $J$ = 8.4 Hz, 1H, H7), 6.77 (dd, $J$ = 7.6, 1.1 Hz, 1H, H5′), 7.04 (d, $J$ = 2.12 Hz, 1H, H4), 7.09 (ddd, $J$ = 11.3, 9.1, 7.0 Hz, 3H, H6′, H3″, H5″), 7.18 (dd, $J$ = 8.4, 2.1 Hz, 1H, H6), 7.47–7.51 (m, 2H, H2″, H6″). $^{13}$C NMR (CD$_3$OD, 150 MHz): $\delta$ ppm 28.3 (C4′), 38.7 (C3′), 42.8 (–CH$_2$–Ar), 62.3 (C3/C1′), 110.0 (C7), 112.3 (C7′), 114.9/115.0 (2C, C3″, C5″), 120.1 (C5′), 123.4 (C4), 127.4 (C8′a), 127.7(C6), 128.2 (C6′), 129.0 (3C, C3a, C2″, C6″), 131.7 (C1″), 135.8 (C5), 138.2 (C4′a), 141.8 (C7a), 154.0 (C8′), 163.2 (C4″), 178.6 (C2). **FTMS + cESI:** $m/z$ 409.11 [M + 1]$^+$.

**2k.** Yield, 1.5 g, 74% (brown solid). M.p. 208–210°C. $^1$H NMR (CD$_3$OD, 600 MHz): $\delta$ ppm 2.86 (dt, $J$ = 16.4, 4.4, 1H, H4′a), 2.99–3.07 (m, 1H, H4′b), 3.18 (ddd, $J$ = 12.7, 5.1, 4.2 Hz, 1H, H3′a), 3.85 (ddd, $J$ = 12.7, 9.3, 4.5 Hz 1H, H3′b), 4.86 (d, $J$ = 15.7 Hz, 1H, –CH$_2$–Ar), 4.99 (d, $J$ = 15.7 Hz, 1H, CH$_2$–Ar), 6.23 (d, $J$ =

8.5 Hz, 1H, H8′), 6.48 (dd, $J$ = 8.5, 2.6 Hz, 1H, H7′), 6.67 (d, $J$ = 2.6, 1H, H5′), 6.94 (d, $J$ = 8.4, 1H, H7), 7.07–7.12 (m, 2H, H3″, H5″), 7.19 (d, $J$ = 2.1 Hz, 1H, H4), 7.28 (dd, $J$ = 8.4, 2.3 Hz, 1H, H6), 7.40–7.44 (m, 2H, H2″, H6″). **¹³C NMR** (CD₃OD, 150 MHz): $\delta$ ppm 28.2 (C4′), 38.3 (C3′), 42.4 (–CH₂–Ar), 63.3 (C3/C1′), 110.4 (C7), 113.8 (C7′), 115.1 (2C, C3″, C5″), 115.3 (C5′), 124.2 (C8′a), 124.7 (C4), 126.9 (C8′), 128.4 (C3a), 128.6 (C6), 129.2 (2C, C2″, C6″), 131.9 (C1″), 136.5 (C5), 137.4 (C4′a), 141.3 (C7a), 156.5 (C6′), 163.2 (C4″), 178.8 (C2). **FTMS + cESI:** $m/z$ 409.11 [M + 1]⁺.

## 3.17. Synthesis of 5-chloro-1-(4-chlorobenzyl)-8′-hydroxy-3′,4′-dihydro-2′H-spiro[indoline-3,1′-isoquinolin]-2-one (1l) and 5-chloro-1-(4-chlorobenzyl)-6′-hydroxy-3′,4′-dihydro-2′H-spiro[indoline-3,1′-isoquinolin]-2-one (2l)

Method D. Prepared from 5-chloro-1-(4-chlorobenzyl)indoline-2,3-dione, **8i** (2.4 g, 9.5 mmol) and 3-hydroxyphenethylamine (1.3 g, 5.0 mmol). The reaction products **1l** and **2l** were separated by column chromatography (hexane : ethyl acetate—80 : 20).

**1l.** Yield, 1.1 g, 35% (brown solid). M.p. 122–124°C. **¹H NMR** (CD₃OD, 600 MHz): $\delta$ ppm 2.97 (m, H4′a), 3.25 (m, 1H, H4′b), 3.36 (dt, $J$ = 13.2, 5.3 Hz, 2H, H3′a, H3′b), 4.80 (d, $J$ = 16.1 Hz, 1H, –CH₂–Ar), 5.15 (d, $J$ = 16.1 Hz, 1H, CH₂–Ar), 6.56 (dd, $J$ = 8.0, 1.1 Hz, 1H, H7′), 6.74 (d, $J$ = 8.4 Hz, 1H, H7), 6.77 (dd, $J$ = 7.6, 1.1 Hz, 1H, H5′), 7.05 (d, $J$ = 2.12 Hz, 1H, H4), 7.10 (t, $J$ = 7.8 Hz, 1H, H6′), 7.18 (dd, $J$ = 8.4, 2.1 Hz, 1H, H6), 7.35–7.37 (d, $J$ = 8.5 Hz, 2H, H3″, H5″), 7.45–7.47 (m, 2H, H2″, H6″). **¹³C NMR** (CD₃OD, 150 MHz): $\delta$ ppm 28.3 (C4′), 38.7 (C3′), 42.9 (–CH₂–Ar), 62.3 (C3/C1′), 110.0 (C7), 112.3 (C7′), 120.1 (C5′), 123.4 (C4), 127.4 (C8′a), 127.8 (C6), 128.2 (C6′), 128.3 (2C, C3″, C5″), 128.7 (2C, C2″, C6″), 132.9 (C4″), 134.7 (C1″), 135.7 (C5), 138.2 (C4′a), 141.8 (C7a), 154.0 (C8′), 178.6 (C2). **FTMS + cESI:** $m/z$ 425.08 [M + 1]⁺.

**2l.** Yield, 2.0 g, 62% (brown solid). M.p. 208–210°C. **¹H NMR** (CD₃OD, 600 MHz): $\delta$ ppm 2.86 (dt, $J$ = 16.4, 4.4, 1H, H4′a), 3.01–3.07 (m, 1H, H4′b), 3.18 (ddd, $J$ = 12.7, 5.5, 4.2 Hz, 1H, H3′a), 3.85 (ddd, $J$ = 12.7, 9.4, 4.2 Hz, 1H, H3′b), 4.98 (d, $J$ = 15.8 Hz, 1H, –CH₂–Ar), 4.98 (d, $J$ = 15.8 Hz, 1H, CH₂–Ar), 6.25 (d, $J$ = 8.5 Hz, 1H, H8′), 6.49 (dd, $J$ = 8.5, 2.6 Hz, 1H, H7′), 6.67 (d, $J$ = 2.6 Hz, 1H, H5′), 6.92 (d, $J$ = 8.5 Hz, 1H, H7), 7.19 (d, $J$ = 2.1 Hz, 1H, H4), 7.28 (dd, $J$ = 8.4, 2.3 Hz, 1H, H6), 7.36–7.40 (m, 4H, H2″ H3″, H5″, H6″). **¹³C NMR** (CD₃OD, 150 MHz): $\delta$ ppm 28.2 (C4′), 38.2 (C3′), 42.5 (–CH₂–Ar), 63.3 (C3/C1′), 110.4 (C7), 113.8 (C7′), 115.3 (C5′), 124.2 (C8′a), 124.7 (C4), 126.9 (C8′), 128.5 (C3a), 128.6 (C6), 128.7 (2C, C2″, C6″), 128.8 (2C, C3″, C5″), 133.2 (C4″), 134.6 (C1″), 136.5 (C5), 137.4 (C4′a), 141.3 (C7a), 156.6 (C6′), 178.8 (C2). **FTMS + cESI:** $m/z$ 425.08 [M + 1]⁺.

## 3.18. Synthesis of 5-chloro-6′-hydroxy-1-(4-methylbenzyl)-3′,4′-dihydro-2′H-spiro[indoline-3,1′-isoquinolin]-2-one (2o)

Method D. Prepared from 5-chloro-1-(4-methylbenzyl)indoline-2,3-dione, **8l** (2 g, 7 mmol), and 3-hydroxyphenethylamine (1.4 g, 10.5 mmol). The reaction product **2o** was purified by column chromatography (hexane : ethyl acetate—70 : 30).

**2o.** Yield, 0.7 g, 25% (brown solid). M.p. 103–105°C. **¹H NMR** (CD₃OD, 600 MHz): $\delta$ ppm 2.33 (s, 3H, Ar–CH₃), 2.86 (dt, $J$ = 16.4, 4.4, 1H, H4′a), 3.03 (ddd, $J$ = 15.3, 9.2, 5.5 Hz 1H, H4′b), 3.15–3.20 (m, 1H, H3′a), 3.84 (ddd, $J$ = 12.8, 9.2, 4.5 Hz 1H, H3′b), 4.79 (d, $J$ = 15.6 Hz, 1H, –CH₂–Ar), 4.99 (d, $J$ = 15.6 Hz, 1H, CH₂–Ar), 6.24 (d, $J$ = 8.5 Hz, 1H, H8′), 6.48 (dd, $J$ = 8.5, 2.6 Hz, 1H, H7′), 6.66 (d, $J$ = 2.6 Hz, 1H, H5′), 6.90 (d, $J$ = 8.5, 1H, H7), 7.16–7.19 (m, 3H, H4, H3″; H5″), 7.23–7.28 (m, 3H, H6, H2″, H6″). **¹³C NMR** (CD₃OD, 150 MHz): $\delta$ ppm 19.8 (Ar – CH₃), 28.3 (C4′), 38.3 (C3′), 42.9 (–CH₂–Ar), 63.4 (C3/C1′), 110.6 (C7), 113.8 (C7′), 115.1 (C5′), 124.3 (C8′a), 124.6 (C4), 127.0 (C8′), 127.2 (2C, C2″, C6″), 128.3 (C3a), 128.6 (C6), 129.1 (2C, C3″, C5″), 132.7 (C1″), 136.5 (C5), 137.4 (C4″), 137.4 (C4′a), 141.5 (C7a), 156.5 (C6′), 178.8 (C2). **FTMS + cESI:** $m/z$ 405.14 [M + 1]⁺.

## 3.19. General method for the synthesis of 6′,7′-dihydroxy-3′,4′-dihydro-2′H-spiro[indoline-3,1′-isoquinolin]-2-ones (3a, 3b) (method E)

Dopamine hydrochloride (1 equiv) and the appropriate isatin (1 equiv) were dissolved in ethanol (10 ml) and triethylamine (1 ml) added (scheme 2c). The reaction mixture was stirred for 72 h at room temperature. At the end of the reaction, the solvent was removed *in vacuo* and distilled water added to the viscous mass. The precipitate obtained was filtered, washed repeatedly with water and dried.

## 3.20. 6′,7′-Dihydroxy-3′,4′-dihydro-2′H-spiro[indoline-3,1′-isoquinolin]-2-one (3a)

Method E. Prepared from isatin (0.4 g, 2.6 mmol) and dopamine (0.5 g, 2.6 mmol). Yield, 0.35 g, 50% (black solid). M.p. 235–237°C. $^1$**H NMR** (CD$_3$OD, 600 MHz): $\delta$ ppm 2.79 (dt, $J = 16.2$, 4.9 Hz, 1H, H4′a), 2.93 (ddd, $J = 16.2$, 8.6, 5.3 Hz, 1H, H4′b), 3.19–3.23 (m, 1H, H3′a), 3.78 (ddd, $J = 13.1$, 8.6, 4.8 Hz, 1H, H3′b), 5.98 (s, 1H, H8′), 6.61 (s, 1H, H5′), 7.00 (d, $J = 7.8$ Hz, 1H, H7), 7.03 (td, $J = 7.6$, 1.1 Hz, 1H, H5), 7.18 (dd, $J = 7.6$, 1.2 Hz, 1H, H4), 7.29 (td, $J = 7.7$ Hz, 1.3 Hz, 1H, H6). $^{13}$**C NMR** (CD$_3$OD, 150 MHz): $\delta$ ppm 27.1 (C4′), 38.5 (C3′), 63.6 (C3/C1′), 109.8 (C7), 112.2 (C8′), 115.2 (C5′), 122.7 (C5), 124.2 (C8′a), 124.6 (C4), 126.9 (C4′a), 129.0 (C6), 134.8 (C3a), 141.8 (C7a), 143.8 (C7′), 144.8 (C6′), 180.9 (C2). **FTMS + cESI:** $m/z$ 283.11 [M + 1]$^+$.

## 3.21. 5-Chloro-6′,7′-dihydroxy-3′,4′-dihydro-2′H-spiro[indoline-3,1′-isoquinolin]-2-one (3b)

Method E. Prepared from 5-chloroisatin (0.5 g, 2.6 mmol) and dopamine (0.5 g, 2.6 mmol). Yield, 0.82 g, 98% (black solid). M.p. 210–213°C. $^1$**H NMR** (CD$_3$OD, 600 MHz): $\delta$ ppm 2.76 (dt, $J = 16.1$, 4.7 Hz, 1H, H4′a), 2.92 (ddd, $J = 16.2$, 8.9, 5.4 Hz, 1H, H4′b), 3.15 (dd, $J = 12.7$, 5.1 Hz, 1H, H3′a), 3.76 (ddd, $J = 13.1$, 8.9, 4.7 Hz, 1H, H3′b), 5.97(s, 1H, H8′), 6.62 (s, 1H, H5′), 6.97 (d, $J = 8.3$ Hz, H7), 7.17 (dd, $J = 1.2$ Hz, 1H, H4), 7.29 (dd, $J = 8.3$, 2.2 Hz, 1H, H6). $^{13}$**C NMR** (CD$_3$OD, 150 MHz): $\delta$ ppm 29.2 (C4′), 38.5 (C3′), 63.8 (C3/C1′), 110.9 (C7), 112.1 (C8′), 115. 3 (C5′), 123.8 (C8′a), 124.9 (C4), 127.1 (C4′a), 127.1 (C5), 128.7 (C6), 137.0 (C3a), 140.6 (C7a), 143.8 (C7′), 145.0 (C6′), 180.6 (C2). **FTMS + cESI:** $m/z$ 317.07 [M + 1]$^+$.

## 3.22. Synthesis of substituted β-nitrostyrenes (10a, 10b) (method F)

The method of Maresh et al. [46] was employed here with some modification (scheme 1c). A solution of substituted benzaldehyde (1 equiv), nitromethane (2 equiv) and anhydrous ammonium acetate (0.1 equiv) in acetic acid was refluxed for 6 h. The reaction mixture was then diluted slowly while stirring, with 200 ml water. During this time, a heavy yellow crystalline mass was formed. This was removed by filtration, washed with water and sucked as dry as possible, recrystallized from boiling methanol and air dried, yielding the β-nitrostyrene as bright yellow crystals in over 95% yield. The compound was used without further purification.

## 3.23. Synthesis of methoxyphenethylamines (11a, 11b) (method F)

This reaction carried out following the method of Maresh et al. [46] (scheme 1c). Typically, 1.0 mmol of nitrostyrene required 2 ml of methanol, 800 mg of zinc dust (12 mmol) and 2 ml of 37% HCl (24 mmol). Methanol was stirred vigorously in an ice bath maintained between 0 and 5°C (ice/NaCl). Conc. HCl, zinc dust and the appropriate nitrostyrene were slowly added over the course of 30 min in small portions while maintaining the temperature between 0 and 5°C. After complete addition of all starting materials, any solids on the side were washed into the solution with a small amount of methanol. The reaction mixture was further stirred for 1 h until the disappearance of the initial yellow colour of the nitrostyrene was observed. The reaction was further stirred for 4–6 h after the yellow colour had disappeared. Thereafter, the reaction mixture was placed in a 4°C refrigerator overnight. Upon completion of the reaction, the excess solid zinc was removed by filtration through the filter paper. The solution was made basic by dropwise addition of saturated sodium hydroxide in methanol, while maintaining the temperature below 5°C, until the pH was greater than 11 as indicated by the pH paper. The product was then extracted into dichloromethane and the solution was dried over anhydrous sodium sulfate. The insoluble zinc hydroxide which precipitates out during the addition of sodium hydroxide solution was extracted many more times with dichloromethane and filtered. The combined organic extracts were evaporated in vacuo to yield the corresponding phenethylamine as a yellow viscous oil in about 70% yield.

## 3.24. General method for the synthesis of 6′-methoxy-3′,4′-dihydro-2′H-spiro[indoline-3,1′-isoquinolin]-2-ones (4a–c, 5a–c, 6a–c) and 1-(substitutedbenzyl)-6′-methoxy-3′,4′-dihydro-2′H-spiro[indoline-3,1′-isoquinolin]-2-ones (4d,e; 5d–f; 6d,e) (method G)

A mixture of the appropriate isatin (1 equiv), 3-methoxyphenethylamine (1.2 equiv) and polyphosphoric acid (2 g) was heated in an oil bath (bath temperature at 100°C), while stirring mechanically for 5 h

(scheme 2*d*). Upon completion of the reaction, as revealed by TLC, the reaction mixture was allowed to cool to about 50°C and quenched by slow addition of water. To this mixture was added a saturated solution of sodium carbonate to make the mixture basic to pH 11. The floating product obtained was extracted into ethyl acetate (3 × 30 ml). The combined organic extracts were dried over anhydrous sodium sulfate, and concentrated under reduced pressure to obtain the crude product. This crude product was purified by flash chromatography on silica gel using suitable solvent systems. Yields ranged between 60 and 98%.

## 3.25. 6′-Methoxy-3′,4′-dihydro-2′H-spiro[indoline-3,1′-isoquinolin]-2-one (**4a**)

Method G. Prepared from isatin (0.81 g, 5.5 mmol), 3-methoxyphenethylamine (1 g, 6.6 mmol) and polyphosphoric acid (3 g). The crude product was purified by flash chromatography (hexane : ethyl acetate—60 : 40). Yield, 1.1 g, 73% (yellow oil). **$^1$H NMR** (CD$_3$OD, 600 MHz): $\delta$ ppm 2.91 (dt, $J$ = 16.6, 4.8 Hz, 1H, H4′a), 3.05 (ddd, $J$ = 16.6, 8.7, 5.4 Hz, 1H, H4′b), 3.20 (dt, $J$ = 12.9, 5.2 Hz, 1H, H3′a), 3.76 (m, 4H, OC$\underline{H}_3$, H3′b), 6.45 (d, $J$ = 8.6 Hz, 1H, H8′), 6.61 (dd, $J$ = 8.6, 2.7 Hz, 1H, H7′), 6.77 (d, $J$ = 2.7 Hz, 1H, H5′), 6.98–7.03 (m, 2H, H5, H7), 7.12–7.14 (m, 1H, H4), 7.29 (td, $J$ = 7.7, 1.3 Hz, 1H, H6). **$^{13}$C NMR** (CD$_3$OD, 150 MHz): $\delta$ ppm 28.4 (C4′), 38.2 (C3′), 54.2 (O$\underline{C}$H$_3$), 63.7 (C3/C1′), 109.7 (C7), 112.6 (C7′), 113.4 (C5′), 122.6 (C5), 124.4 (C4), 126.1 (C8′a), 127.2 (C8′), 128.9 (C6), 135.2 (C3a), 137.3 (C4′a), 141.8 (C7a), 158.8 (C6′), 181.1 (C2). **FTMS + cESI:** $m/z$ 281.13 [M + 1]$^+$.

## 3.26. 5-Chloro-6′-methoxy-3′,4′-dihydro-2′H-spiro[indoline-3,1′-isoquinolin]-2-one (**4b**)

Method G. Prepared from 5-chloroisatin (1 g, 5.5 mmol), 3-methoxyphenethylamine (1 g, 6.6 mmol) and polyphosphoric acid (5 g). The crude product was purified by flash chromatography (hexane : ethyl acetate—50 : 50). Yield, 0.6 g, 35% (green solid). M.p. 114–116°C. **$^1$H NMR** (CD$_3$OD, 600 MHz): $\delta$ ppm 2.90 (dt, $J$ = 16.6, 4.6 Hz, 1H, H4′a), 3.05 (ddd, $J$ = 16.5, 8.9, 5.4 Hz, 1H, H4′b), 3.17 (dt, $J$ = 12.7, 5.0 Hz, 1H, H3′a), 3.77 (m, 4H, OC$\underline{H}_3$, H3′b), 6.46 (d, $J$ = 8.6 Hz, 1H, H8′), 6.65 (dd, $J$ = 8.7, 2.7 Hz, 1H, H7′), 6.78 (d, $J$ = 2.7 Hz, 1H, H5′), 6.97 (d, $J$ = 8.3 Hz, 1H, H7), 7.13 (d, $J$ = 2.1 Hz, 1H, H4), 7.30 (dd, $J$ = 8.3, 2.2 Hz, 1H, H6), **$^{13}$C NMR** (CD$_3$OD, 150 MHz): $\delta$ ppm 28.4 (C4′), 38.2 (C3′), 54.3 (O$\underline{C}$H$_3$), 63.8 (C3/C1′), 110.9 (C7), 112.7 (C7′), 113.5 (C5′), 124.8 (C4), 125.5 (C8′), 127.0 (C8′), 127.6 (C3a), 128.7 (C6), 137.1 (C5), 137.4 (C4′a), 140.6 (C7a), 159.0 (C6′), 180.7 (C2). **FTMS + cESI:** $m/z$ 315.09 [M + 1]$^+$.

## 3.27. 5,7-Dibromo-6′-methoxy-3′,4′-dihydro-2′H-spiro[indoline-3,1′-isoquinolin]-2-one (**4c**)

Method G. Prepared from 5,7-dibromoisatin, **7c** (2.02 g, 6–6 mmol), 3-methoxyphenethylamine (1 g, 6.6 mmol) and polyphosphoric acid (3 g). The crude product was purified by flash chromatography (hexane : ethyl acetate—90 : 10). Yield 2.03 g, 70% (brown solid). M.p. 255–257°C. **$^1$H NMR** (CD$_3$OD, 600 MHz): $\delta$ ppm 3.52 (dt, $J$ = 16.1, 3.9 Hz, 1H, H4′a), 3.73 (ddd, $J$ = 15.4, 9.5, 5.5 Hz, 1H, H4′b), 3.79 (ddd, $J$ = 12.2, 5.5, 3.6 Hz, 1H, H3′a), 4.40 (m, 1H, H3′b), 4.52 (s, 3H, OC$\underline{H}_3$), 7.43 (dd, $J$ = 8.6, 2.8 Hz, 1H, H7′), 7.56 (d, $J$ = 2.8 Hz, 1H, H5′), 7.94 (d, $J$ = 1.8 1H, H4), 7.18 (d, $J$ = 8.6 Hz, 1H, H8′), 8.48 (d, $J$ = 1.9 Hz, 1H, H6), 11.57 (br. s, 1H, H1). **$^{13}$C NMR** (CD$_3$OD, 150 MHz): $\delta$ ppm 30.1 (C4′), 39.1 (C3′), 56.3 (O$\underline{C}$H$_3$), 65.6 (C3/C1′), 104.1 (C7), 114.1 (C7′), 115.0 (C5′), 115.2 (C5), 127.4 (C8′a), 127.7 (C4), 128.3 (C8′), 134.5 (C6), 139.1 (C4′a), 140.7 (C3a), 142.7 (C7a), 159.3 (C6′), 181.0 (C2). **FTMS + cESI:** $m/z$ 438.95 [M + 1]$^+$.

## 3.28. 6′-Methoxy-1-(4-fluorobenzyl)-3′,4′-dihydro-2′H-spiro[indoline-3,1′-isoquinolin]-2-one (**4d**)

Method G. Prepared from 1-(4-fluorobenzyl)indoline-2,3-dione, **8a** (1.5 g, 6.0 mmol), 3-methoxyphenethylamine (0.9 g, 6.0 mmol) and polyphosphoric acid (3 g). The crude product was purified by flash chromatography (hexane : ethyl acetate—70 : 30). Yield, 1.7 g, 73% (yellow oil). **$^1$H NMR** (CD$_3$OD, 600 MHz): $\delta$ ppm 2.93 (dt, $J$ = 16.5, 4.5 Hz, 1H, H4′a), 3.06–3.11 (m, 1H, H4′b), 3.22 (dt, $J$ = 12.8, 5.0 Hz, 1H, H3′a), 3.76 (s, 3H, 6′-OC$\underline{H}_3$), 3.87 (ddd, $J$ = 13.3, 9.1, 4.6 Hz, 1H, H3′b), 4.90 (d, $J$ = 15.6, 1H, –C$\underline{H}_2$–Ar), 5.00 (d, $J$ = 15.6 Hz, 1H, C$\underline{H}_2$–Ar), 6.32 (d, $J$ = 8.6 Hz, 1H, H8′), 6.59 (dd, $J$ = 8.6, 2.7 Hz, 1H, H7′), 6.79 (d, $J$ = 2.7 Hz, 1H, H5′), 6.97 (d, $J$ = 8.0 Hz, 1H, H7), 7.05 (td, $J$ = 7.5, 0.9 Hz, 1H, H5) 7.07–7.11 (m, 2H, H3″,H5″), 7.18 (dd, $J$ = 7.4, 1.2 Hz, 1H, H4), 7.28 (td, $J$ = 7.8, 1.3 Hz, 1H, H6),

7.42–7.46 (m, 2H, H2″, H6″). **$^{13}$C NMR** (CD$_3$OD, 150 MHz): δ ppm 28.4 (C4′), 38.3 (C3′), 42.4 (–$\underline{C}$H$_2$–Ar), 54.1 (O$\underline{C}$H$_3$), 63.3 (C3/C1′), 109.3 (C7), 112.6 (C7′), 113.6 (C5′), 115.0 (2C, C3″,C5″), 123.2 (C5), 124.3 (C4), 126.1 (C8′a), 127.0 (C8′), 128.9 (C6), 129.2 (2C, C2″,C6″), 132.1 (C1″), 134.5 (C3a), 137.5 (C4′a), 142.6 (C7a), 158.9 (C6′), 163.2 (C4″), 179.0 (C2). **FTMS + cESI:** *m/z* 389.17 [M + 1]$^+$.

## 3.29. 6′-Methoxy-1-(4-methylbenzyl)-3′,4′-dihydro-2′H-spiro[indoline-3,1′-isoquinolin]-2-one (**4e**)

Method G. Prepared from **8e** (1.5 g, 6.0 mmol), 3-methoxyphenethylamine (0.9 g, 0.9 ml, 6.0 mmol) and polyphosphoric acid (3 g), and purified by flash chromatography (hexane : ethyl acetate—70 : 30). Yield, 2.0 g, 87% (yellow oil). **$^1$H NMR** (CD$_3$OD, 600 MHz): δ ppm 2.32 (s, 3H, 4″-$\underline{C}$H$_3$–Bz), 2.94 (dt, *J* = 16.5, 4.6 Hz, 1H, H4′a), 3.06–3.12 (m, 1H, H4′b), 3.21–3.26 (m, 1H, H3′a), 3.75 (s, 3H, 6′-O$\underline{C}$H$_3$), 3.88 (ddd, *J* = 13.3, 9.0, 4.7 Hz, 1H, H3′b), 4.81 (d, J = 15.5, 1H, –$\underline{C}$H$_2$–Ar), 5.00 (d, *J* = 15.5 Hz, 1H, $\underline{C}$H$_2$–Ar), 6.34 (d, *J* = 8.7 Hz, 1H, H8′), 6.58 (dd, *J* = 8.6, 2.7 Hz, 1H, H7′), 6.79 (d, *J* = 2.7 Hz, 1H, H5′), 6.95 (d, *J* = 7.9 Hz, 1H, H7), 7.03 (td, *J* = 7.5, 1.0 Hz, 1H, H5) 7.15–7.18 (m, 3H, H4, H3″,H5″), 7.25 (td, *J* = 7.7, 1.3 Hz, 1H, H6), 7.28 (d, *J* = 7.9 Hz, 2H, H2″,H6″). **$^{13}$C NMR** (CD$_3$OD, 150 MHz): δ ppm 19.8 (4″–$\underline{C}$H$_3$), 28.3 (C4′), 38.3 (C3′), 42.9 (–$\underline{C}$H$_2$–Ar), 54.3 (O$\underline{C}$H$_3$), 63.3 (C3/C1′), 109.5 (C7), 112.7 (C7′), 113.6 (C5′), 123.2 (C5), 124.2 (C4), 126.0 (C8′a), 127.1 (C8′), 127.2 (2C, C2″, C6″), 128.9 (C6), 129.1 (2C, C3″, C5″), 133.0 (C1″), 134.2 (C3a), 137.2 (C4″), 137.3 (C4′a), 142.7 (C7a), 158.9 (C6′), 178.8 (C2). **FTMS + cESI:** *m/z* 385.19 [M + 1]$^+$.

## 3.30. 6′,7′-Dimethoxy-3′,4′-dihydro-2′H-spiro[indoline-3,1′-isoquinolin]-2-one (**5a**)

Method G. Prepared from isatin (1.0 g, 6.8 mmol), 3,4-dimethoxyphenethylamine, **11a**, (1 g, 6.6 mmol) and polyphosphoric acid (3 g). The crude product was purified by flash chromatography (hexane : ethyl acetate—70 : 30). Yield, 2.07 g, 99% (yellow solid). M.p. 188–191°C. **$^1$H NMR** (CD$_3$OD, 600 MHz): δ ppm 2.86 (dt, *J* = 16.3, 4.9 Hz, 1H, H4′a), 2.95–3.03 (m, 1H, H4′b), 3.21 (dt, *J* = 12.9, 5.2 Hz, 1H, H3′a), 3.52 (s, 3H, 7′-O$\underline{C}$H$_3$), 3.76 (ddd, *J* = 13.1, 8.5, 4.8 Hz, 1H, H3′b), 3.83 (s, 3H, 6′-O$\underline{C}$H$_3$), 6.03 (s, 1H, H8′), 6.79 (s, 1H, H5′), 7.01 (dt, *J* = 7.8, 0.8 Hz, H7), 7.03 (td, *J* = 7.6, 1.1 Hz, 1H, H5), 7.15 (ddd, *J* = 7.5, 1,3, 0.6 Hz, 1H, H4), 7.30 (td, *J* = 7.7, 1.3 Hz, 1H, H6). **$^{13}$C NMR** (CD$_3$OD, 150 MHz): δ ppm 27.6 (C4′), 38.4 (C3′), 54.9 (7′-O$\underline{C}$H$_3$), 55.0 (6′-O$\underline{C}$H$_3$), 63.8 (C3/C1′), 109.2 (C8′), 109.8 (C7), 112.2 (C5′), 122.6 (C5), 124.5 (C4), 125.8 (C8′a), 128.9 (C4′a), 129.0 (C6), 135.0 (C3a), 141.7 (C7a), 147.7 (C7′), 148.0 (C6′), 180.9 (C2). **FTMS + cESI:** *m/z* 311.14 [M + 1]$^+$.

## 3.31. 5-Chloro-6′,7′-dimethoxy-3′,4′-dihydro-2′H-spiro[indoline-3,1′-isoquinolin]-2-one (**5b**)

Method G. From 5-chloroisatin (1.5 g, 8.3 mmol), 3,4-dimethoxyphenethylamine, **11a** (1.5 g, 8.3 mmol) and polyphosphoric acid (3 g). The crude product was purified by flash chromatography (hexane : ethyl acetate— 70 : 30). Yield, 1.8 g, 63% (white solid). M.p. 138–140°C. **$^1$H NMR** (CD$_3$OD, 600 MHz): δ ppm 2.83–2.88 (m, 1H, H4′a), 2.96–3.02 (m, 1H, H4′b), 3.18 (dt, *J* = 12.7, 5.1 Hz, 1H, H3′a), 3.56 (s, 3H, 7′-O$\underline{C}$H$_3$), 3.75 (ddd, *J* = 13.1, 8.6, 4.7 Hz, 1H, H3′b), 3.84 (s, 3H, 6′-O$\underline{C}$H$_3$), 6.03 (s, 1H, H8′), 6.81 (s, 1H, H5′), 6.99 (dt, *J* = 8.3, 1H, H7), 7.16 (d, *J* = 2.1 Hz, 1H, H4), 7.30–7.32 (m, 1H, H6). **$^{13}$C NMR** (CD$_3$OD, 150 MHz): δ ppm 27.5 (C4′), 38.3 (C3′), 55.0 (7′-O$\underline{C}$H$_3$), 55.1 (6′-O$\underline{C}$H$_3$), 64.0 (C3/C1′), 109.0 (C8′), 111.0 (C7), 112.3 (C5′), 124.8 (C4), 125.1 (C8′a), 127.7 (C3a), 128.9 (C6), 129.0 (C4′a), 136.8 (C5), 140.6 (C7a), 147.8 (C7′), 148.9 (C6′), 180.6 (C2). **FTMS + cESI:** *m/z* 345.10 [M + 1]$^+$.

## 3.32. 5,7-dibromo-6′,7′-dimethoxy-3′,4′-dihydro-2′H-spiro[indoline-3,1′-isoquinolin]-2-one (**5c**)

Method G. Prepared from 5,7-dibromoisatin, **7c** (1.0 g, 3.3 mmol), 3,4-dimethoxyphenethylamine, **11a** (0.6 g, 3.3 mmol) and polyphosphoric acid (3 g). The crude product was purified by flash chromatography (hexane : ethyl acetate—70 : 30). Yield, 1.8 g, 63% (white solid). M.p. 180–183°C. **$^1$H NMR** (CD$_3$OD D, 600 MHz): δ ppm 2.84 (dt, *J* = 16.3, 4.7, 1H, H4′a), 2.96–3.03 (m, 1H, H4′b), 3.16 (dt, *J* = 12.8, 5.1 Hz, 1H, H3′a), 3.58 (s, 3H, 7′-O$\underline{C}$H$_3$), 3.73 (ddd, *J* = 13.1, 8.7, 4.7 Hz, 1H, H3′b), 3.84 (s, 3H, 6′-O$\underline{C}$H$_3$), 6.04 (s, 1H, H8′), 6.81 (s, 1H, H5′), 7.26 (d, *J* = 1.8 Hz, 1H, H4), 7.66 (d, *J* = 1.8 1H, H6). **$^{13}$C NMR** (CD$_3$OD, 150 MHz): δ ppm 27.5 (C4′), 38.2 (C3′), 55.0 (7′-O$\underline{C}$H$_3$), 55.2 (6′-O$\underline{C}$H$_3$), 64.9

(C3/C1′), 103.0 (C7), 109.0 (C8′), 112.4 (C5′), 114.9 (C5), 124.6 (C8′a), 126.7 (C4), 129.1 (C4′a), 133.9 (C6), 138.2 (C5), 141.0 (C7a), 147.9 (C7′), 149.1 (C6′), 179.7 (C2). **FTMS + cESI:** $m/z$ 468.96 $[M + 1]^+$.

## 3.33. 1-(4-Fluorobenzyl)-6′,7′-dimethoxy-3′,4′-dihydro-2′H-spiro[indoline-3,1′-isoquinolin]-2-one (5d)

Method G. Prepared from 1-(4-fluorobenzyl)indoline-2,3-dione, **8a** (2.0 g, 7.8 mmol), 3,4-dimethoxyphenethylamine, **11a** (1.4 g, 7.8 mmol) and polyphosphoric acid (2.5 g). The crude product was purified by flash chromatography (hexane : ethyl acetate—90 : 10). Yield, 2.2 g, 88% (brown solid). M.p. 145–149°C. **$^1$H NMR** (CD$_3$OD, 600 MHz): δ ppm 2.88 (dt, $J$ = 16.2, 4.6 Hz, 1H, H4′a), 2.99–3.06 (m, 1H, H4′b), 3.21–3.26 (m, 1H, H3′a), 3.38 (s, 3H, 7′-OCH$_3$), 3.82 (s, 4H, H3′b, 6′-OCH$_3$), 4.82 (d, $J$ = 15.5 Hz, 1H, –CH$_2$–Ar), 5.09 (d, $J$ = 15.5 Hz, 1H, CH$_2$–Ar), 5.82 (s, 1H, H8′), 6.80 (s, 1H, H5′), 7.04–7.11 (m, 4H, H5, H7, H3″,H5″), 7.20 (dd, $J$ = 7.5, 1.2 Hz, 1H, H4), 7.31 (td, $J$ = 7.8, 1.3 Hz, 1H, H6), 7.45–7.48 (m, 2H, H2″, H6″). **$^{13}$C NMR** (CD$_3$OD, 150 MHz): δ ppm 27.6 (C4′), 38.5 (C3′), 42.3 (–CH$_2$–Ar), 54.8 (7′-OCH$_3$), 55.0 (6′-OCH$_3$), 63.3 (C3/C1′), 108.8 (C8′), 109.3 (C7), 112.2 (C5′), 115.2 (2C, C3″,C5″), 123.2 (C5), 124.3 (C4), 125.8 (C8′a), 128.9 (C6), 129.4 (2C, C2″,C6″), 132.4 (C1″), 134.3 (C3a), 137.4 (C4″), 142.4 (C7a), 147.7 (C7′), 148.8 (C6′), 163.2 (C4″), 178.8 (C2). **FTMS + cESI:** $m/z$ 419.18 $[M + 1]^+$.

## 3.34. 6′,7′-Dimethoxy-1-(4-methylbenzyl)-3′,4′-dihydro-2′H-spiro[indoline-3,1′-isoquinolin]-2-one (5e)

Method G. Prepared from 1-(4-methylbenzyl)indoline-2,3-dione, **8e** (1.5 g, 6.0 mmol), 3,4-dimethoxyphenethylamine, **11a** (1 g 6.0 mmol) and polyphosphoric acid (2.5 g). Purified by flash chromatography (hexane : ethyl acetate—90 : 10). Yield, 2.2 g, 88% (brown solid). M.p. 175–177°C. **$^1$H NMR** (CD$_3$OD, 600 MHz): δ ppm 2.32 (s, 3H, 4″-CH$_3$–Bz), 2.89 (dt, $J$ = 16.2, 4.7 Hz, 1H, H4′a), 3.03 (ddd, $J$ = 16.2, 8.7, 5.3 Hz, 1H, H4′b), 3.24 (dt, $J$ = 12.8, 5.1 Hz, 1H, H3′a), 3.37 (s, 3H, 7′-OCH$_3$), 3.83 (s, 4H, H3′b, 6′-OCH$_3$), 4.75 (d, $J$ = 15.4 Hz, 1H, –CH$_2$–Ar), 5.13 (d, $J$ = 15.4 Hz, 1H, CH$_2$–Ar), 5.82 (s, 1H, H8′), 6.81 (s, 1H, H5′), 7.04 (d, $J$ = 7.9 Hz, 1H, H7), 7.06 (td, $J$ = 7.6, 1.0 Hz, 1H, H5), 7.17 (d, $J$ = 7.9 Hz, 2H, H3″, H5″), 7.20 (dd, $J$ = 7.5, 1.2 Hz, 1H, H4), 7.30 (td, $J$ = 7.8, 1.3 Hz, 1H, H6), 7.32 (m, 2H, H2″, H6″). **$^{13}$C NMR** (CD$_3$OD, 150 MHz): δ ppm 19.7 (4″-CH$_3$), 27.5 (C4′), 38.5 (C3′), 42.8 (–CH$_2$–Ar), 54.7 (7′-OCH$_3$), 55.0 (6′-OCH$_3$), 63.4 (C3/C1′), 108.8 (C8′), 109.5 (C7), 112.2 (C5′), 123.2 (C5), 124.2 (C4), 125.8 (C8′a), 127.4 (2C, C2″, C6″), 128.8 (C4′a), 128.9 (C6), 129.0 (2C, C3″,C5″), 133.2 (C1″), 134.3 (C3a), 137.4 (C4″), 142.6 (C7a), 147.7 (C7′), 148.7 (C6′), 178.9 (C2). **FTMS + cESI:** $m/z$ 415.20 $[M + 1]^+$.

## 3.35. 1-(4-Bromobenzyl)-5-chloro-6′,7′-dimethoxy-3′,4′-dihydro-2′H-spiro[indoline-3,1′-isoquinolin]-2-one (5f)

Method G. Prepared from 1-(4-bromobenzyl)-5-chloroindoline-2,3-dione, **8c** (0.7 g, 4.1 mmol), 3,4-dimethoxyphenethylamine, **11a** (1.2 g 3.4 mmol) and polyphosphoric acid (2.0 g). Product purified by flash chromatography (hexane : ethyl acetate—80 : 20). Yield, 1.2 g, 69% (brown solid). M.p. 198–201°C. **$^1$H NMR** (CD$_3$OD, 600 MHz): δ ppm 2.84–2.90 (m, 1H, H4′a), 3.02 (ddd, $J$ = 9.7, 5.3, 4,5 Hz, 1H, H4′b), 3.18–3.23 (m, 1H, H3′a), 3.42 (s, 3H, 7′-OCH$_3$), 3.82 (d, $J$ = 5.9 Hz, 1H, H3′b), 3.84 (s, 3H, 6′-OCH$_3$), 4.83 (d, $J$ = 15.7 Hz, 1H, –CH$_2$–Ar), 5.06 (d, $J$ = 15.7 Hz, 1H, CH$_2$–Ar), 5.80 (s, 1H, H8′), 6.82 (s, 1H, H5′), 7.04 (d, $J$ = 8.4 Hz, 1H, H7), 7.21(d, $J$ = 72.2 Hz, 1H, H4), 7.33 (dd, $J$ = 8.4, 2.2 Hz, 1H, H6), 7.37 (d, $J$ = 8.4 Hz, 2H, H2″, H6″), 7.50–7.54 (m, 2H, H3″, H5″). **$^{13}$C NMR** (CD$_3$OD, 150 MHz): δ ppm 27.5 (C4′), 38.4 (C3′), 42.5 (–CH$_2$–Ar), 54.9 (7′-OCH$_3$), 55.0 (6′-OCH$_3$), 63.4 (C3/C1′), 108.6 (C8′), 110.5 (C7), 112.4 (C5′), 121.4 (C4″), 124.7 (C4), 125.0 (C8′a), 127.8 (C4′a), 128.5 (C6), 129.0 (C5), 129.4 (2C, C2″, C6″), 131.6 (2C, C3″,C5″), 135.3 (C1″), 136.2 (C3a), 141.1 (C7a), 147.8 (C7′), 149.0 (C6′), 178.5 (C2). **FTMS + cESI:** $m/z$ 513.06 $[M + 1]^+$.

## 3.36. 6′,7′,8′-Trimethoxy-3′,4′-dihydro-2′H-spiro[indoline-3,1′-isoquinolin]-2-one (6a)

Method G. Prepared from isatin (1.6 g, 10.9 mmol) and 3,4,5-trimethoxyphenethylamine, **11b** (2.8 g, 13 mmol), and polyphosphoric acid (3 g). The crude product was purified by flash chromatography (hexane : ethyl acetate—80 : 20). Yield, 1.24 g, 34% (white solid). M.p. 238–240°C. **$^1$H NMR** (CD$_3$OD, 600 MHz): δ ppm 2.85–2.97 (m, 2H, H4′a, H4′b), 3.21 (s, 3H,8′-OCH$_3$), 3.26–3.30 (m, 2H, H3′a, H3′b),

3.69 (s, 3H, 7'-OC$\underline{H}_3$), 3.86 (s, 3H, 6'-OC$\underline{H}_3$), 6.65 (s, 1H, H5'), 6.96 (dt, $J$ = 7.8, 0.8 Hz, H7), 6.99 (dt, td, $J$ = 7.8, 0.8 Hz, 1H, H4), 7.06 (ddd, $J$ = 7.4, 1.2, 0.6 Hz, 1H, H5), 7.25 (td, $J$ = 7.7, 1.3 Hz, 1H, H6). $^{13}$C NMR (CD$_3$OD, 150 MHz): $\delta$ ppm 28.0 (C4'), 38.6 (C3'), 55.0 (6'-O$\underline{C}$H$_3$), 58.5 (8'-O$\underline{C}$H$_3$), 59.5 (7'-O$\underline{C}$H$_3$), 62.4 (C3/C1'), 107.5 (C5'), 109.7 (C7), 120.4 (C8'a), 121.7 (C4), 121.7 (C5), 128.4 (C6), 132.5 (C4'a), 135.1 (C3a), 140.0 (C7a), 142.2 (C6'), 150.1 (C8'), 153.2 (C7'), 181.0 (C2). **FTMS + cESI:** $m/z$ 341.15 [M + 1]$^+$.

## 3.37. 5-Chloro-6',7',8'-trimethoxy-3',4'-dihydro-2'H-spiro[indoline-3,1'-isoquinolin]-2-one (**6b**)

Method G. Prepared from 5-chloroisatin (1.5 g, 8.3 mmol), 3,4,5-trimethoxyphenethylamine, **11b** (2.1 g, 9.9 mmol) and polyphosphoric acid (3 g). Purified by flash chromatography (hexane : ethyl acetate—80 : 20). Yield, 1.8 g, 58% (brown solid). M.p. 102–105°C. $^1$H NMR (CD$_3$OD, 600 MHz): $\delta$ ppm 2.86–2.98 (m, 2H, H4'a, H4'b), 3.18–3.24 (m, 1H, H3'a), 3.28–3.32 (m, 4H, H3'b, 8'-OC$\underline{H}_3$), 3.70 (s, 3H, 7'-OC$\underline{H}_3$), 3.87 (s, 3H, 6'-OC$\underline{H}_3$), 6.65 (s, 1H, H5'), 6.96 (dt, $J$ = 8.3, 1H, H7), 7.05 (d, td, $J$ = 2.1 Hz, 1H, H4), 7.26 (dd $J$ = 8.3, 2.1 Hz, 1H, H6).$^{13}$C NMR (CD$_3$OD, 150 MHz): $\delta$ ppm 27.9 (C4'), 38.5 (C3'), 55.1 (6'-O$\underline{C}$H$_3$), 58.6 (8'-O$\underline{C}$H$_3$), 59.5 (7'-O$\underline{C}$H$_3$), 62.4 (C3/C1'), 107.5 (C5'), 110.7 (C7), 119.6 (C8'a), 123.9 (C4), 126.8 (C5), 128.2 (C6), 132.6 (C4'a), 137.0 (C3a), 140.0 (C8'), 140.9 (C7a), 150.0 (C6'), 153.4 (C7'), 181.0 (C2). **FTMS + cESI:** $m/z$ 375.11 [M + 1]$^+$.

## 3.38. 5,7-Dibromo-6',7',8'-trimethoxy-3',4'-dihydro-2'H-spiro[indoline-3,1'-isoquinolin]-2-one (**6c**)

Method G. Prepared from 5,7-dibromoisatin, **7c**, (1.5 g, 5.0 mmol), 3,4,5-trimethoxyphenethylamine, **11b** (1.3 g, 6 mmol) and polyphosphoric acid (2 g). The crude product was purified by flash chromatography (hexane : ethyl acetate—80 : 20). Yield, 2.1 g, 84% (brown solid). M.p. 200–203°C. $^1$H NMR (CD$_3$OD, 600 MHz): $\delta$ ppm 2.89 (td, $J$ = 6.5, 5.5, 2H, H4'a, H4'b), 3.18 (ddd, $J$ = 13.1, 7.4, 5.6, 1H, H3'a,), 3.27–3.31 (m, 1H, H3'b), 3.37 (s, 8'-OC$\underline{H}_3$), 3.70 (s, 3H, 7'-OC$\underline{H}_3$), 3.87 (s, 3H, 6'-OC$\underline{H}_3$), 6.65 (s, 1H, H5'), 7.15 (d, $J$ = 1,8 Hz, 1H, H4), 7.60 (d, $J$ = 1.8, 1H, H6). $^{13}$C NMR (CD$_3$OD, 150 MHz): $\delta$ ppm 27.8 (C4'), 38.4 (C3'), 55.1 (6'-O$\underline{C}$H$_3$), 58.6 (8'-O$\underline{C}$H$_3$), 59.5 (7'-O$\underline{C}$H$_3$), 63.3 (C3/C1'), 102.8 (C7), 107.5 (C5'), 114.0 (C5), 119.2 (C8'a), 125.6 (C4), 132.7 (C4'a), 133.1 (C6), 138.4 (C3a), 139.9 (C8'), 141.2 (C7a), 150.0 (C6'), 153.6 (C7'), 179.8 (C2). **FTMS + cESI:** $m/z$ 498.97 [M + 1]$^+$.

## 3.39. 1-(4-Fluorobenzyl)-6',7',8'-trimethoxy-3',4'-dihydro-2'H-spiro[indoline-3,1'-isoquinolin]-2-one (6d)

Method G. Prepared from 1-(4-fluorobenzyl)indoline-2,3-dione, **8a** (2.0 g, 7.8 mmol), 3,4,5-trimethoxyphenethylamine, **11b** (1.7 g, 7.8 mmol) and polyphosphoric acid (2.5 g). The crude product was purified by flash chromatography (hexane : ethyl acetate—80 : 20). Yield, 3.0 g, 62% (white solid). M.p. 173–175°C. $^1$H NMR (CD$_3$OD, 600 MHz): $\delta$ ppm 2.87 (s, 3H, 8'-OC$\underline{H}_3$), 2.89–3.0 (m,, 2H, H4'a, H4'b), 3.27 (ddd, $J$ = 13.2, 7.7, 5.1 Hz, 1H, H3'a), 3.36 (m. 1H, H3'b), 3.70 (s, 3H, 7'-OC$\underline{H}_3$), 3.87 (s, 6'-OC$\underline{H}_3$), 4.91 (d, $J$ = 15.6 Hz, 1H, –C$\underline{H}_2$–Ar), 5.08 (d, $J$ = 15.6 Hz, 1H, C$\underline{H}_2$–Ar), 6.67 (s, 1H, H5'), 6.95 (dt, $J$ = 7.8, 0.7 Hz, 1H, H7), 7.01 (td, $J$ = 7.5, 1.0 Hz, 1H, H5), 7.08–7.13 (m, 3H, H4, H3'', H5''), 7.25 (td, $J$ = 7.8, 1.3 Hz, 1H, H6), 7.54–7.58 (m, 2H, H2'', H6''). $^{13}$C NMR (CD$_3$OD, 150 MHz): $\delta$ ppm 28.1 (C4'), 38.7 (C3'), 42.7 (–C$\underline{H}_2$–Ar), 55.1 (6'-O$\underline{C}$H$_3$), 58.3 (8'-O$\underline{C}$H$_3$),59.5 (7'-O$\underline{C}$H$_3$) 62.0 (C3/C1'), 107.6 (C5'), 109.2 (C7), 114.9 (2C, C3'', C5''), 120.0 (C8'a), 122.4 (C4), 123.4 (C5), 128.4 (C6), 127.6 (2C, C2'', C6''), 132.2 (C3a), 132.8 (C4'a), 134.7 (C1''), 140.1 (C8'), 142.8 (C7a), 150.0 (C7'), 153.3 (C6'), 163.2 (C4''), 178.9 (C2). **FTMS + cESI:** $m/z$ 449.19 [M + 1]$^+$.

## 3.40. 6',7',8'-Trimethoxy-1-(4-methylbenzyl)-3',4'-dihydro-2'H-spiro[indoline-3,1'-isoquinolin]-2-one (6e)

Method G. Prepared from 1-(4-methylbenzyl)indoline-2,3-dione, **8e** (1.6 g, 6.4 mmol), 3,4,5-trimethoxyphenethylamine, **11b** (1.6 g, 6 mmol) and polyphosphoric acid (2 g). The crude product was purified by flash chromatography (hexane : ethyl acetate—80 : 20). Yield, 2.3 g, 80% (brown solid). M.p. 135–138°C. $^1$H NMR (CD$_3$OD, 600 MHz): $\delta$ ppm 2.34 (s, 3H, 4''-C$\underline{H}_3$–Bz), 2.89 (s, 3H, 8'-OC$\underline{H}_3$), 2.90–2.96 (m, 2H, H4'a, H4'b), 3.24–3.30 (m, 1H, H3'a), 3.34–3.37 (m, 1H, H3'b), 3.71 (s, 3H, 7'-OC$\underline{H}_3$), 3.87 (s, 4H, H3'b, 6'-OC$\underline{H}_3$), 4.87 (d, $J$ = 15.5 Hz, 1H, –C$\underline{H}_2$–Ar), 5.06 (d, $J$ = 15.5 Hz, 1H, C$\underline{H}_2$–Ar), 6.67

(s, 1H, H5′), 6.92 (d, *J* = 7.9 Hz, 1H, H7), 6.99 (td, *J* = 7.6, 1.0 Hz, 1H, H5), 7.11 (dd, *J* = 7.4, 1.2 Hz, 1H, H4), 7.17–7.20 (m, 2H, H3″, H5″), 7.22 (td, *J* = 7.8, 1.3 Hz, 1H, H6), 7.40–7.42 (m, 2H, H2″, H6″). **¹³C NMR** (CD₃OD, 150 MHz): $\delta$ ppm 19.8 (4″-$\underline{C}H_3$), 28.1 (C4′), 38.7 (C3′), 43.3 (–$\underline{C}H_2$–Ar), 55.1 (6′-O$\underline{C}H_3$), 58.3 (8′-O$\underline{C}H_3$), 59.5 (7′-O$\underline{C}H_3$) 62.0 (C3/C1′), 107.5 (C5′), 109.4 (C7), 120.1 (C8′a), 122.3 (C5), 123.3 (C4), 127.6 (2C, C2″,C6″), 128.8 (2C, C3″,C5″), 132.7 (C4′a), 133.1 (C3a), 134.1 (C1″), 137.1 (C4″), 140.1 (C8′), 143.0 (C7a), 150.1 (C7′), 153.2 (C6′), 178.9 (C2). **FTMS + cESI:** *m/z* 445.21 [M + 1]⁺.

Data accessibility. Electronic supplementary material available: ¹H NMR and ¹³C NMR spectra of target compounds, S2–S81; LC-MS Spectra of target compounds, S82–121; biological Screening Data, electronic supplementary material, S122–127.

Authors' contributions. S.M.N.E. conceived and designed the project, subsequently supervised the work; reviewed the data and prepared the manuscript. M.M.M.L. carried out the synthesis of the compounds, performed the interpretation of spectral data and co-wrote the manuscript.

Competing interests. We declare we have no competing interests.

Funding. Funding was provided by the Ministry of Higher Education of Cameroon under the Research Modernization Scheme.

Acknowledgements. The authors are indebted to Prof. Michael Spiteller, INFU, Technische Universität Dortmund, Germany, for providing all of the HPLC-MS, ¹³C and ¹NMR data for this study. Anti-cancer screening was carried out by the US National Cancer Institute, Bethesda, MD under the NCI-60 screening programme.

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
