## [Reviewer comments · Royal Society Open Science]

Review History

RSOS-191316.R0 (Original submission)

Review form: Reviewer 1

Is the manuscript scientifically sound in its present form?

No

Are the interpretations and conclusions justified by the results?

No

Is the language acceptable?

Yes

Do you have any ethical concerns with this paper?

No

Have you any concerns about statistical analyses in this paper?

Yes

Recommendation?

Major revision is needed (please make suggestions in comments)

Comments to the Author(s)

In the present study, twenty-four 3',4'-dihydro-2'H-spiro[indolin-3:1'-isoquinolin]-2-ones, designed as molecular hybrids of tetrahydroisoquinolines and oxindole, were synthesized and screened in vitro against 59 cell lines in the NCI 60 screen. Among the synthesized compounds, few of them showed an antiproliferative activity with IC₅₀ values around ten micromolar. For the most active compounds, the IC₅₀ values should be included. A brief discussion of the structure activity relationship (SAR) around this series of compounds should be discussed. Also the mechanism of action of the most active compounds of the series should be reported

Review form: Reviewer 2**Is the manuscript scientifically sound in its present form?**

Yes

Are the interpretations and conclusions justified by the results?

Yes

Is the language acceptable?

Yes

Do you have any ethical concerns with this paper?

No

Have you any concerns about statistical analyses in this paper?

No

Recommendation?

Accept with minor revision (please list in comments)

Comments to the Author(s)

This paper reports the synthesis of 3',4'-dihydro-2'H-spiro[indoline-3:1'-isoquinolin]-2-ones and the evaluation of their antiproliferative activity. The authors make great play of the biological activity of molecular hybrids of THIQ and oxindole, this work does in my view merit publication in Royal Society Open Science. My detailed comments are as followed:

In table 1, actually three compounds(3b, 4d, 5f) inhibited cell proliferation by at least 50%. I think it is not rigorous to say that four compounds displayed at least 50% inhibition of cell proliferation in this paper.

In the experimental part, I think it would help if methods A, B, C et al were defined in scheme 1 and scheme 2.

I do think that it is necessary for a paper to unify the written format and typesetting, such as the written format of "FTMS+cESI" of 1c is different from others'. And the layout of the NMR, MS data for compounds is also not uniform.

Decision letter (RSOS-191316.R0)

15-Oct-2019

Dear Dr Efange:

Title: 3',4'-Dihydro-2'H-spiro[indolin-3:1'-isoquinolin]-2-ones as Potential Anticancer Agents:
Synthesis and Preliminary Screening
Manuscript ID: RSOS-191316

The editor assigned to your manuscript has now received comments from reviewers. We would like you to revise your paper in accordance with the referee and Subject Editor suggestions which can be found below (not including confidential reports to the Editor). Please note this decision does not guarantee eventual acceptance.

Please submit your revised paper before 07-Nov-2019. Please note that the revision deadline will expire at 00.00am on this date. If we do not hear from you within this time then it will be assumed that the paper has been withdrawn. In exceptional circumstances, extensions may be possible if agreed with the Editorial Office in advance. We do not allow multiple rounds of revision so we urge you to make every effort to fully address all of the comments at this stage. If deemed necessary by the Editors, your manuscript will be sent back to one or more of the original reviewers for assessment. If the original reviewers are not available we may invite new reviewers.

Please also include the following statements alongside the other end statements. As we cannot publish your manuscript without these end statements included, if you feel that a given heading is not relevant to your paper, please nevertheless include the heading and explicitly state that it is not relevant to your work.

- Acknowledgements

- Funding statement

Please include a funding section after your main text which lists the source of funding for each author.

Once again, thank you for submitting your manuscript to Royal Society Open Science and I look

forward to receiving your revision. If you have any questions at all, please do not hesitate to get in touch.

On behalf of the Subject Editor Professor Anthony Stace and the Associate Editor Dr Andrew Harned.

RSC Associate Editor:

Comments to the Author:

In this manuscript the authors report on the synthesis and biological evaluation of a series of spirooxindoles. Even though the antiproliferative activities were, ultimately, rather low, the work as a whole appears to be hypothesis driven. As such it is suitable for publication in this journal. Nevertheless, the reviewers have identified a few deficiencies that should be addressed. In particular, the authors should craft a summary of the SAR data that could potentially be used to develop more active compounds in this series. They should also ensure that the introductory text provides a strong argument as to why this compounds are of interest (provide a strong hypothesis).

RSC Subject Editor:

Comments to the Author:

(There are no comments.)

Reviewers' Comments to Author:

Reviewer: 1

Comments to the Author(s)

In the present study, twenty-four 3',4'-dihydro-2'H-spiro[indolin-3:1'-isoquinolin]-2-ones, designed as molecular hybrids of tetrahydroisoquinolines and oxindole, were synthesized and screened in vitro against 59 cell lines in the NCI 60 screen. Among the synthesized compounds, few of them showed an antiproliferative activity with IC50 values around ten micromolar. For the most active compounds, the IC50 values should be included. A brief discussion of the structure activity relationship (SAR) around this series of compounds should be discussed. Also the mechanism of action of the most active compounds of the series should be reported

Reviewer: 2

Comments to the Author(s)

This paper reports the synthesis of 3',4'-dihydro-2'H-spiro[indoline-3:1'-isoquinolin]-2-ones and the evaluation of their antiproliferative activity. The authors make great play of the biological

activity of molecular hybrids of THIQ and oxindole, this work does in my view merit publication in Royal Society Open Science. My detailed comments are as followed:

In table 1, actually three compounds(3b, 4d, 5f) inhibited cell proliferation by at least 50%. I think it is not rigorous to say that four compounds displayed at least 50% inhibition of cell proliferation in this paper.

In the experimental part, I think it would help if methods A, B, C et al were defined in scheme 1 and scheme 2.

I do think that it is necessary for a paper to unify the written format and typesetting, such as the written format of "FTMS+cESI" of 1c is different from others'. And the layout of the NMR, MS data for compounds is also not uniform.

Author's Response to Decision Letter for (RSOS-191316.R0)

See Appendix A.

Decision letter (RSOS-191316.R1)

19-Nov-2019

Dear Dr Efang:

Title: 3',4'-Dihydro-2'H-spiro[indolin-3:1'-isoquinolin]-2-ones as Potential Anticancer Agents: Synthesis and Preliminary Screening
Manuscript ID: RSOS-191316.R1

It is a pleasure to accept your manuscript in its current form for publication in Royal Society Open Science. The chemistry content of Royal Society Open Science is published in collaboration with the Royal Society of Chemistry.

Royal Society of Chemistry
Thomas Graham House
Science Park, Milton Road
Cambridge, CB4 0WF

Royal Society Open Science - Chemistry Editorial Office

On behalf of the Subject Editor Professor Anthony Stace and the Associate Editor Dr Andrew Harned.

RSC Associate Editor

Comments to the Author:

The authors have done a good job responding to the questions and concerns raised by the previous review. I agree that a more detailed SAR conclusion is probably premature at this time. Likewise, given the nature of this study (proof-of-principle), and the author's reliance on others for biological testing, I don't feel IC50 values are needed at this time. In its current form, this work will likely provide inspiration for others looking for new scaffolds for drug discovery/design.

Reviewer(s)' Comments to Author:

Appendix A

October 31, 2019

The Publishing Editor, Journals

Royal Society of Chemistry

Thomas Graham House

Science Park, Milton Road

Cambridge, CB4 0WF

Royal Society Open Science – Chemistry Editorial Office

RE: MANUSCRIPT ID RSOS-191316

Dear Sir/Ms:

We write to acknowledge receipt of your letter of 15th October, 2019 which provides details of the review of our manuscript. We thank the reviewers for their thoughtful comments. In a bid to strengthen the work, we have provided our response to these comments point-by-point below. We trust you will find that the responses clearly address the issues raised by the reviewers and we look forward to a favorable ruling on the revised manuscript.

Sincerely,

S. Mbua Ngale Efang, Ph.D.

Professor of Chemistry, University of Buea

RESPONSES TO REFEREES

RSC Associate Editor

Summary of SAR data: A summary of the SAR data has been inserted (see response to Reviewer 1 below).

Hypothesis:

The underlying hypothesis of this work is presented on page 5, lines 13-20, and also copied here:

“By extension, spirooxindoles, specifically 3’4’-dihydro-2’H-spiro[indoline-3:1’-isoquinoline]-2-ones, that are obtained from the Pictet-Spengler type reaction between phenethylamines with isatin, have also received scant attention. Formally, the latter spirooxindoles can be regarded as molecular hybrids of THIQ and OX arising from the Pictet-Spengler type reaction between a substituted phenethylamine and OX. In view of the antiproliferative activity of both THIQs and oxindoles (including spirooxindoles), we proposed that this particular class of spirofused molecular hybrids of THIQ and OX would display antiproliferative activity.”

Acknowledgements:

These have been inserted as requested (see page 43).

Funding Statement

The Funding Statement has been inserted in the revised manuscript on page 43.

REVIEWER 1

Inclusion of IC50 values: Our laboratory at the University of Buea in Cameroon is not equipped to carry out screening on cancer cells. Therefore, we rely solely on the National Cancer Institute, Bethesda, MD, USA, which provided these preliminary results. Under the NCI60 protocol, compounds that display growth inhibition equal to or greater than 50% are usually selected for secondary screening to obtain a dose response curve and IC50 value. Unfortunately, even though three compounds displayed 50% inhibition of cell proliferation, they were not subjected to secondary screening, possibly because the values are too close to the cut-off point. Therefore, we cannot provide IC50 values for these compounds at this time. However, as we embark on optimization studies, we hope to discover more potent analogues for which IC50 values will be provided.

Structure-Activity Relationships: In view of the relatively small number of compounds synthesized and tested, and given the large number of cell lines in the screen, it not clear that one can obtain a robust SAR from this data set. However, some trends are apparent particularly for some lines such as the renal cancer cells (A498 and CAKI-1) which appear to be most susceptible to this class of compounds (see Table 1). Therefore, we have provided the following brief SAR summary (page 8, para 2 & page 9, para 1) which provides the way forward for future work:

“Among the spirooxindoles showing evidence of antiproliferative activity, analogues containing the 6,7-dimethoxy-THIQ fragment (series 5) appeared with the highest frequency while those containing the 6-methoxy-THIQ fragment (series 3) appeared with the lowest frequency. Therefore, the 6,7-dimethoxy-THIQ fragment appears to be the preferred fragment for subsequent optimization efforts of this hybrid scaffold. Of the twenty compounds showing noticeable antiproliferative activity nine (45%) were unsubstituted at the nitrogen atom of the oxindole fragment, while the rest featured a substituted benzyl fragment at this position. Therefore, subsequent optimization efforts may concentrate on the synthesis of either N-unsubstituted or N-substituted analogues. In light of these findings, we conclude that the 3'4'-dihydro-2'H-spiro[indoline-3:1'-isoquinolin]-2-one scaffold does possess antiproliferative activity.”

Mechanism of action: As to the mechanism of action, no studies have been conducted at this time. However, we note that the anticancer activity of the isomeric 2',4'-dihydro-1'H-spiro[indoline-3:3'-isoquinolin]-2-ones has attributed to inhibition of Ras-GTP. The following segment has been inserted into the manuscript (page 9, para 2) to address the reviewer's comment:

“A recent investigation of the isomeric 2',4'-dihydro-1'H-spiro[indoline-3:3'-isoquinolin]-2-ones, published while the current study was underway, found that the latter compounds display anticancer activity on colon cancer.⁵¹ The most potent compound of that study was found to inhibit Ras-GTP and thereby suppress downstream signaling by MAPK, PI3K-Akt and Wnt ultimately resulting in mitochondrial apoptosis.⁵¹ Therefore, while the mode of action of the target 3',4'-dihydro-2'H-spiro[indoline-3:1'-isoquinolin]-2-ones of the present study is unknown, it does appear that spirofused molecular hybrids of THIQ and OX may be potentially useful leads for anticancer drug discovery.”

REVIEWER 2

Table 1 and the number compounds showing activity greater than 50%: We have corrected the text to show that three (and not four) compounds were found to display at least 50% inhibition of cancer cell proliferation. The correction appears in the Abstract (line 6) and Results & Discussion (page 8, para 1, line 1).

Definition of Methods A-G, in Scheme 1 and Scheme 2: The two schemes have been modified to clearly identify the synthetic methods employed for the preparation of each set of compounds.

Uniformity of reporting format and typesetting: The reporting format and typesetting has been modified to provide a uniform presentation of NMR and MS data.